# Sageflow: Robust Federated Learning against Both Stragglers and Adversaries

**Jungwuk Park**[*]
KAIST
savertm@kaist.ac.kr

**Dong-Jun Han**[*]
KAIST
djhan93@kaist.ac.kr

**Minseok Choi**
Jeju National University
ejaqmf@jejunu.ac.kr

**Jaekyun Moon**
KAIST
jmoon@kaist.edu

## Abstract

While federated learning (FL) allows efficient model training with local data at edge devices, among major issues still to be resolved are: slow devices known as stragglers and malicious attacks launched by adversaries. While the presence of both of these issues raises serious concerns in practical FL systems, no known schemes or combinations of schemes effectively address them at the same time. We propose Sageflow, staleness-aware grouping with entropy-based filtering and loss-weighted averaging, to handle both stragglers and adversaries simultaneously. Model grouping and weighting according to staleness (arrival delay) provides robustness against stragglers, while entropy-based filtering and loss-weighted averaging, working in a highly complementary fashion at each grouping stage, counter a wide range of adversary attacks. A theoretical bound is established to provide key insights into the convergence behavior of Sageflow. Extensive experimental results show that Sageflow outperforms various existing methods aiming to handle stragglers/adversaries.

## 1 Introduction

Large volumes of data collected at various edge devices (i.e., smart phones) are valuable resources in training a model with a good performance. Federated learning [18, 12, 7] is a promising direction for large-scale learning, which enables training of a global model with less privacy concerns. However, among major issues that need to be addressed in current federated learning (FL) systems are the devices called stragglers that are considerably slower than the average and the adversaries that enforce a various form of attacks.

Regarding the first issue, simply waiting for all the stragglers at each global round can significantly slow down the overall training process. To handle stragglers, asynchronous FL schemes [15, 22, 25, 21, 14] update the global model every time the server receives a local model from each device; especially in FedAsync [25], the global model is updated asynchronously according to the device's *staleness* $t - \tau$, the time difference between the current round $t$ and the past round $\tau$ when the device first received the global model. While the asynchronous schemes are highly effective in handling stragglers, the one-by-one update nature does not lend itself well for integration with established aggregation methods to combat the second issue, the adversaries.

There are different forms of adversarial attacks that significantly degrade current FL systems. In untargeted attacks, an attacker can poison the updated model at the devices before it is sent to the

---

[*]Equal contribution.

35th Conference on Neural Information Processing Systems (NeurIPS 2021).

server (model poisoning) [3, 9] or can poison the datasets of each device (data poisoning) [2, 16]. In targeted attacks (or backdoor attacks) [4, 1, 20], the adversaries cause the model to misclassify the targeted subtasks only, while not degrading the overall test accuracy that much. Robust federated averaging (RFA) of [19], a well-known method proposed to handle adversaries in FL, employs geometric-median-based aggregation to provide a fair level of protection. Other various aggregation schemes (e.g., Multi-Krum) also successfully handle adversaries in distributed learning [3, 28, 5]. Unfortunately, however, the performance of these methods are substantially degraded when the portion of adversaries is large. The presence of stragglers can drive the attack ratio higher (e.g., by ignoring stragglers), significantly degrading the performance of current aggregation schemes.

While the presence of both stragglers and adversaries raises significant concerns in practical FL, to our knowledge, there are currently no existing methods or known combinations of ideas that can effectively handle these two issues simultaneously.

**Main contributions.** We propose Sageflow, staleness-aware grouping with entropy-based filtering and loss-weighted averaging, a robust FL strategy which can handle both stragglers and adversaries at the same time. Targeting the straggler issue, our strategy is to perform periodic global aggregation while allowing the results sent from stragglers to be aggregated in later rounds. In each global round, we take advantage of the results sent from stragglers by first grouping the models that come from the same initial models (i.e., same staleness), to obtain a group representative model. Then, we aggregate the representative models of all groups based on their staleness, to obtain the global model. Our periodic aggregation strategy is not only effective in neutralizing stragglers but also provides a great platform for countering adversaries, as discussed below.

Targeting each grouping stage of our straggler-mitigating idea, we propose an intra-group defense strategy which is based on our entropy-based filtering and loss-weighted averaging. The entropy filtering first filters out the models with high entropies, i.e., outliers, and the loss-weighted averaging of the survived models enables model aggregation according to the measured qualities of the received local models. These two methods work in a highly complementary fashion to effectively counter a wide range of adversarial attacks in each grouping stage. Here, in computing the entropy and loss of each received model, we utilize *public data* that we assume to be available at the server. In fact, the utilization of public data is not a new idea, as seen in recent FL setups of [30, 27, 11]. This is generally a reasonable setup since data centers typically have some collected data that can be accessed by public. For example, different types of anonymous medical data are often available for public research in various countries. We show later via experiments that only a very small amount of public data is necessary at the server (1-2% of the entire dataset, which is comparable to the amount of local data at a single device) to successfully combat adversaries. Our main contributions are as follows:

- We propose **Sageflow**, handling both stragglers/adversaries simultaneously in FL, via a novel **staleness-aware grouping** combined with **entropy filtering and loss-weighted averaging**.

- We derive the **theoretical bound** for Sageflow based on key parameters and provide insights into the convergence behavior.

- Experimental results on different datasets show that Sageflow **outperforms various combinations of straggler/adversary defense methods** using only a small portion of public data.

**Related works.** The authors of [15, 22, 25, 17, 21] have recently tackled the straggler issue in FL. The basic idea is to allow the devices and the server to update the models asynchronously; the global model is updated every time the server receives a model from a device. However, a critical issue is that grouping-based (or aggregation-based) methods designed to handle adversaries, such as RFA [19], Multi-Krum [3] or the presently proposed entropy/loss based idea, are hard to be implemented in conjunction with these schemes since the model update is performed one-by-one asynchronously. Compared to the asynchronous schemes, our staleness-aware grouping can be combined smoothly with various aggregation rules against adversaries; we can apply RFA, Multi-Krum or our entropy/loss based idea in each grouping stage to obtain the group representative model.

To combat adversaries, various aggregation methods have been proposed in a distributed learning setup with IID (independent, identically distributed) data across nodes [28, 29, 5, 3, 24]. The authors of [5] suggests a geometric-median-based aggregation rule of models or gradients. In [28], a trimmed mean approach is proposed which removes a fraction of largest and smallest values of each element among the received results. In Multi-Krum [3], among $N$ workers in the system, the server tolerates $f$ Byzantine workers, where $2f + 2 < N$. Targeting FL with non-IID data distribution, RFA of [19] utilizes the geometric median of models sent from devices, similar to [5]. However, as

already implied, these methods are ineffective when combined with a straggler-mitigation scheme (e.g., ignoring stragglers), potentially degrading the performance of learning. Compared to Multi-Krum and RFA, our entropy/loss based scheme can tolerate adversaries even with a high attack ratio, showing remarkable advantages, especially when combined with straggler-mitigation schemes. Moreover, our approach enables defense against a wider variety of attacks by considering both entropy and loss, which are shown to play highly complementary roles.

Finally, we note that the authors of [26], [27] proposed Zeno and Zeno+ that also utilize public data at the server to handle adversaries, but in a distributed learning setup with IID data across the nodes. Compared to Zeno+, our Sageflow targets non-IID data distribution in a FL setup. While Zeno+ adopts a fully asynchronous update rule (without considering the staleness) with the loss-based score function, our Sageflow integrates staleness-aware grouping, a semi-synchronous straggler-handling method, with entropy filtering and loss-weighted averaging, a harmonized means to provide protection against a wider variety of attacks. We show later via experiments that our Sageflow outperforms Zeno+ in practical FL setups having stragglers and various types of adversaries.

## 2 Proposed Sageflow for Federated Learning

We consider the following federated optimization problem:

$$\mathbf{w}^* = \underset{\mathbf{w}}{\operatorname{argmin}} \ F(\mathbf{w}) = \underset{\mathbf{w}}{\operatorname{argmin}} \sum_{k=1}^{N} \frac{m_k}{m} F_k(\mathbf{w}), \tag{1}$$

where $N$ is the number of devices, $m_k$ is the number of data samples in device $k$, and $m = \sum_{k=1}^{N} m_k$ is the total number of data samples of all $N$ devices in the system. By letting $x_{k,j}$ be the $j$-th data sample in device $k$, the local loss function of device $k$, $F_k(\mathbf{w})$, is written as $F_k(\mathbf{w}) = \frac{1}{m_k} \sum_{j=1}^{m_k} \ell(\mathbf{w}; x_{k,j})$.

### 2.1 Staleness-Aware Grouping against Stragglers

In the $t$-th global round of a practical FL setting, the server sends the current model $\mathbf{w}_t$ to $K$ devices in $S_t$, a randomly selected subset of $N$ devices. We let $C = K/N$ be the ratio of devices that participate in each global round. Each device in $S_t$ performs $E$ local updates with its own data and sends the updated model back to the server.

In handling slow devices, our idea assumes periodic global aggregation at the server. At each global round $t$, the server transmits the current model and time stamp, $(\mathbf{w}_t, t)$, to the devices in $S_t$. Instead of waiting for all devices in $S_t$, the server aggregates the models that arrive by a fixed time deadline $T_d$ to obtain $\mathbf{w}_{t+1}$, and moves on to the next global round $t + 1$. Hence, model aggregation is performed periodically with every $T_d$. A key feature here is that we do not ignore the results sent from stragglers not arriving by the deadline $T_d$. These results are utilized at the next global aggregation step, or even later, depending on the delay or staleness. Let $U_i^{(t)}$ be the group of devices selected from the server at global round $t$ that successfully sent their results to the server at global round $i$ for $i \geq t$. Then, we can write $S_t = \cup_{i=t}^{\infty} U_i^{(t)}$, where $U_i^{(t)} \cap U_j^{(t)} = \emptyset$ for $i \neq j$. Here, $U_\infty^{(t)}$ can be viewed as the devices that are selected at round $t$ but failed to successfully send their results back to the server. According to these notations, the devices whose training results arrive at the server during global round $t$ belong to one of the $t + 1$ groups: $U_t^{(0)}, U_t^{(1)}, ..., U_t^{(t)}$. Note that the result sent from device $k \in U_t^{(i)}$ is the model after $E$ local updates starting from $\mathbf{w}_i$, and we denote this model by $\mathbf{w}_i(k)$. At each round $t$, we first perform FedAvg as

$$\mathbf{v}_{t+1}^{(i)} = \sum_{k \in U_t^{(i)}} \frac{m_k}{\sum_{k \in U_t^{(i)}} m_k} \mathbf{w}_i(k) \tag{2}$$

for all $i = 0, 1, ..., t$. Here, $\mathbf{v}_{t+1}^{(i)}$ can be viewed as a group representative model, which is the aggregated result of locally updated models (starting from $\mathbf{w}_i$) received at round $t$ with staleness $t - i + 1$. Then from the representative models of all $t$ groups, $\mathbf{v}_{t+1}^{(0)}, \mathbf{v}_{t+1}^{(1)}, ..., \mathbf{v}_{t+1}^{(t)}$, we take a weighted averaging according to different staleness: $\sum_{i=0}^{t} \alpha_t^{(i)}(\lambda) \mathbf{v}_{t+1}^{(i)}$. Here, $\alpha_t^{(i)}(\lambda) \propto \frac{\sum_{k \in U_t^{(i)}} m_k}{(t-i+1)^\lambda}$ is a normalized coefficient that is proportional to the number of data samples in $U_t^{(i)}$ and inversely proportional to $(t - i + 1)^\lambda$, for a given staleness exponent $\lambda \geq 0$. Hence, we have a larger weight

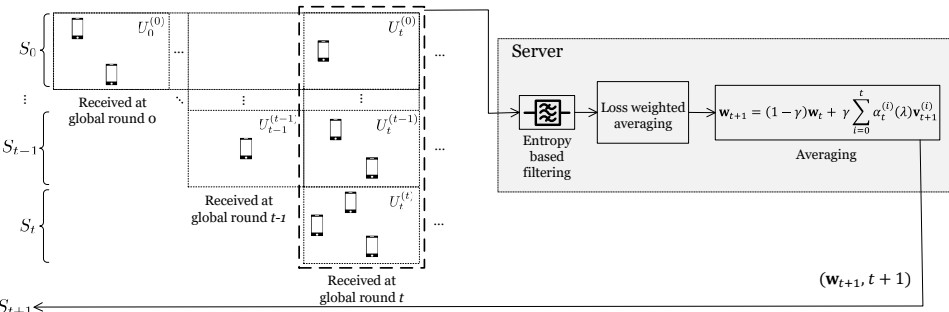

Figure 1: Overall procedure of Sageflow. At global round $t$, each received model belongs to one of the $t+1$ sets: $U_t^{(0)}, U_t^{(1)}, ..., U_t^{(t)}$. After entropy-based filtering, the server performs loss-weighted averaging for the results that belong to $U_t^{(i)}(E_{th})$ to obtain $\mathbf{v}_{t+1}^{(i)}$. Then we aggregate $\{\mathbf{v}_{t+1}^{(i)}\}_{i=0}^t$ with $\mathbf{w}_t$ to obtain $\mathbf{w}_{t+1}$, and move on to the next round $t+1$.

for $\mathbf{v}_{t+1}^{(i)}$ with a smaller $t-i+1$ (staleness). This is to give more weights to more recent results having smaller staleness. Based on the weighted sum $\sum_{i=0}^t \alpha_t^{(i)}(\lambda)\mathbf{v}_{t+1}^{(i)}$, we finally obtain $\mathbf{w}_{t+1}$ as

$$\mathbf{w}_{t+1} = (1-\gamma)\mathbf{w}_t + \gamma \sum_{i=0}^t \alpha_t^{(i)}(\lambda)\mathbf{v}_{t+1}^{(i)}, \tag{3}$$

where $\gamma$ is the time-average coefficient. Now we move on to the next round $t+1$, where the server selects $S_{t+1}$ and sends $(\mathbf{w}_{t+1}, t+1)$ to these devices. Here, if the server knows the set of active devices (which are still performing computation), $S_{t+1}$ can be constructed to be disjoint with the active devices. If not, the server randomly chooses $S_{t+1}$ among all devices in the system and the selected active devices can ignore the current request of the server. The left-hand side of Fig. 1 describes our staleness-aware grouping method.

The key characteristics of our approach against stragglers can be summarized as follows. First, by periodic global aggregation at the server, our scheme is not delayed by the effect of stragglers. Secondly, our scheme fully utilizes the results sent from stragglers in the future global rounds.

## 2.2 Entropy-based Filtering and Loss-Weighted Averaging against Adversaries

In this subsection, we propose a two-stage solution against adversaries which not only shows better performance with attacks but also combines well with our staleness-aware grouping scheme compared to existing aggregation methods to handle adversaries. Our idea is to utilize a small amount of *public data* collected at the server. Data centers typically have their own public data samples as utilized in recent FL setups of [30, 27, 11], e.g., various anonymous medical data that are open for public research. We provide the following two solutions which can protect the system in a highly complementary fashion against various attacks including model poisoning, data poisoning and scaled backdoor attacks using only a very small amount of public data. It is shown later in Section G of Supplementary Material that our idea works well even with a class-imbalanced public data.

**1) Entropy-based filtering.** Let $n_{pub}$ be the number of public data samples in the server. We also let $x_{pub,j}$ be the $j$-th sample in the server. When the server receives the locally updated models from the devices, it measures the entropy of each device $k$ by utilizing the public data as

$$E(k) = \frac{1}{n_{pub}} \sum_{j=1}^{n_{pub}} E_{x_{pub,j}}(k), \tag{4}$$

where $E_{x_{pub,j}}(k)$ is the shannon entropy of the model of the $k$-th device on the sample $x_{pub,j}$ written as $E_{x_{pub,j}}(k) = -\sum_{q=1}^Q P_{x_{pub,j}}^{(q)}(k) \log P_{x_{pub,j}}^{(q)}(k)$. Here, $Q$ is the number of classes of the dataset and $P_{x_{pub,j}}^{(q)}(k)$ is the probability of prediction for the $q$-th class on a sample $x_{pub,j}$, using the model of the $k$-th device. In supervised learning, the model produces a high-confident prediction for the ground truth labels of the trained samples and thus has a low entropy for the prediction. However, as can be seen from Fig. 2(a) with the FMNIST dataset [23], under specific model poisoning attacks (reverse sign attack with scale 0.1 in this case), the models compromised by adversaries tend to

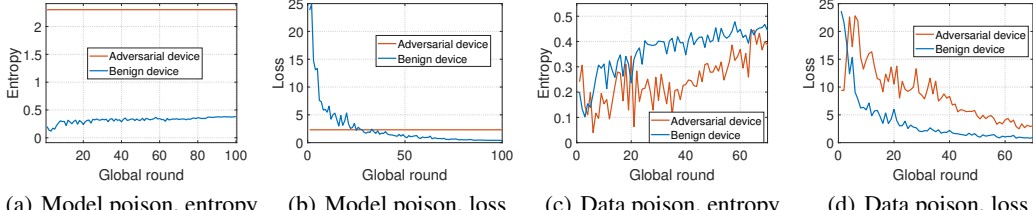

Figure 2: **Model poisoning ((a), (b)):** We can filter out the models of adversaries via entropy, but not via loss. **Data poisoning ((c), (d)):** We can reduce the impact of adversaries via loss, but not via entropy.

predict randomly for all classes and thus have high entropies compared to the models of benign devices. Even when the local dataset of the benign device is biased, it has a high-confident prediction (i.e., low entropy) on the classes that are in their local dataset, making the entropy lower compared to the model of the adversaries (as seen in Fig. 2(a)). Based on this observation, we let the server to filter out the models that have entropies greater than some threshold value $E_{th}$. We note that the inflicted models cannot be filtered out based on the loss values in the setting of model poisoning in Fig. 2 (see Fig. 2(b)), which confirms the importance of the role of entropy. It is shown later in Section 4 that $E_{th}$ is a hyperparameter that can be easily tuned since there is a huge gap between the entropy values of benign versus adversarial devices. We also note that our entropy-based filtering is robust against attacks even with a large portion of adversaries, since it just filters out the results whose entropies are greater than $E_{th}$. This is a significant advantage compared to current aggregation methods against adversaries, whose performances are substantially degraded when the attack ratio is high.

We note that there are some cases where entropy cannot play a role, e.g., when the attacker forces the model to be biased toward specific classes. Data poisoning is one example of such attack. In such cases, the models of the adversaries generally produce high loss values and thus loss-weighted averaging (which will be described shortly) can play a key role.

**2) Loss-weighted averaging.** The server also measures the loss of each model using the public data. Based on the loss values, the server aggregates the models as follows:

$$\mathbf{w}_{t+1} = \sum_{k \in S_t} \beta_t^{(k)}(\delta) \mathbf{w}_t(k), \text{ where} \tag{5}$$

$$\beta_t^{(k)}(\delta) \propto \frac{m_k}{\{F_{pub}(\mathbf{w}_t(k))\}^\delta} \text{ and } \sum_{k \in S_t} \beta_t^{(k)}(\delta) = 1. \tag{6}$$

Here, $\mathbf{w}_t(k)$ is the locally updated model of the $k$-th device at global round $t$. $F_{pub}(\mathbf{w}_t(k))$ is defined as the averaged loss of $\mathbf{w}_t(k)$ computed on public data at the server, i.e., $F_{pub}(\mathbf{w}_t(k)) = \frac{1}{n_{pub}} \sum_{j=1}^{n_{pub}} \ell(\mathbf{w}_t(k); x_{pub,j})$. Finally, $\delta(\geq 0)$ in $\{F_{pub}(\cdot)\}^\delta$ is a loss exponent related to the impact of the loss measured with public data. We note that a setting $\delta = 0$ in (5) reduces our loss-weighted averaging method to FedAvg of (1). Under the data poisoning attack or model replacement backdoor attack (or scaled backdoor attack) in [1], the models of adversaries have relatively larger losses compared to others. Fig. 2(d) shows an example with FMNIST dataset under data poisoning attack. By the definition of $\beta_t^{(k)}(\delta)$, the data-poisoned model would be given a small weight and has a less impact on the next global update. By replacing FedAvg with the above loss-weighted averaging, we are able to build a system that is highly robust against data poisoning and scaled backdoor attacks. As can be seen from Fig. 2(c), the impact of data poisoning cannot be filtered out via entropy measures, since the benign and inflicted models have similar entropy values. This indicates that the loss measure has its own unique role, along with the entropy measure.

Entropy-based filtering and loss-weighted averaging can be easily combined, and work complementarily to tackle model poisoning, data poisoning and scaled backdoor attacks. Utilizing only one of these methods significantly degrade the performance, as shown later in Section F of Supplementary Material.

## 2.3 Sageflow

Finally we put together Sageflow, which can handle both stragglers and adversaries at the same time by applying entropy-based filtering and loss-weighted averaging in each grouping stage of our straggler-mitigating staleness-aware aggregation. The details of overall Sageflow operation are described in Algorithm 1 and Fig. 1.

We stress that Sageflow performs well with only a very small amount of public data at the server, 1% of the entire dataset, which is comparable to the amount of local data at a single device. This would not cause large delay as the server generally has a large computing resources compared to that of devices. The computational complexity of Sageflow depends on the number of received models at each global round, and the running time for computing the entropy/loss of each model. Assuming that the complexity of computing entropy or loss is linear to the number of model parameters as in [26], the time complexity of our scheme can be written as $\mathcal{O}(n_{pub}|w|K)$ (where $|w|$ is the number of model parameters), which scales linearly with $K$, the number of devices. The time complexity of Zeno is the same as ours, but suffers from performance degradation as can be seen in the experimental results in Section 4. Compared to RFA, Sageflow requires higher complexity by the factor $n_{pub}$. This additional computation of Sageflow compared to RFA is the cost for considerably better robustness against adversaries.

---

**Algorithm 1** Proposed Sageflow Algorithm

---

**Input:** Initialized model $\mathbf{w}_0$, **Output:** Final global model $\mathbf{w}_T$
**Process at the Server**
1: **for** each global round $t = 0, 1, ..., T - 1$ **do**
2:     Choose $S_t$ and send the current model and the global round $(\mathbf{w}_t, t)$ to the devices
3:     Wait for $T_d$ and then:
4:     **for** $i = 0, 1, ..., t$ **do**
5:         $U_t^{(i)}(E_{th}) = \{k \in U_t^{(i)} | E(k) < E_{th}\}$     // Entropy-based filtering in each group
6:         $\mathbf{v}_{t+1}^{(i)} = \sum_{k \in U_t^{(i)}(E_{th})} \beta_t^{(k)}(\delta)\mathbf{w}_i(k)$   // Loss-weighted averaging in each group (with same staleness)
7:     **end for**
8:     $\mathbf{w}_{t+1} = (1 - \gamma)\mathbf{w}_t + \gamma \sum_{i=0}^{t} \alpha_t^{(i)}(\lambda)\mathbf{v}_{t+1}^{(i)}$   // Averaging of representative models (with different staleness)
9: **end for**

**Process at the Device**: Device $k$ receives $(\mathbf{w}_t, t)$ from the server and performs local updates to obtain $\mathbf{w}_t(k)$. Then each benign device $k$ sends $(\mathbf{w}_t(k), t)$ to the server, while a malicious adversary sends a poisoned model depending on the type of attack.

---

## 3 Convergence Analysis

In this section, we provide insights into the convergence of Sageflow with the following standard assumptions in FL [13, 25].

**Assumption 1** *The global loss fuction $F$ defined in (1) is $\mu$-strongly convex, i.e., $F(x) \leq F(y) + \nabla F(x)^T(x-y) - \frac{\mu}{2}\|x-y\|^2$ for all $x, y$. Moreover, $F$ is $L$-smooth, i.e., $F(x) \geq F(y) + \nabla F(x)^T(x-y) - \frac{L}{2}\|x - y\|^2$ for all $x, y$.*

**Assumption 2** *Let $\mathbf{w}_t^i(k)$ be the model of the $k$-th benign device after $i$ local updates starting from global round $t$. Let $\xi_t^i(k)$ be a set of data samples that are randomly selected from the device $k$ for $(i+1)$-th local update. Then, $\mathbb{E}\|\nabla F_k(\mathbf{w}_t^i(k), \xi_t^i(k)) - \nabla F(\mathbf{w}_t^i(k))\|^2 \leq \rho_1$ holds for all $t$ and $k = 1, \ldots, N$ and $i = 1, \ldots, E$.*

Let $B_t^{(i)}$ and $M_t^{(i)}$ be the set for benign and adversarial devices of $U_t^{(i)}$ respectively, satisfying $U_t^{(i)} = B_t^{(i)} \cup M_t^{(i)}$ and $B_t^{(i)} \cap M_t^{(i)} = \emptyset$. Similarly, define $B_t^{(i)}(E_{th})$ and $M_t^{(i)}(E_{th})$ as the sets obtained after entropy-based filtering is applied to $B_t^{(i)}$ and $M_t^{(i)}$. Now we have the following definition which describes the effect of the adversaries.

**Definition 1** *Let $\Omega_t^{(i)}(E_{th}, \delta)$ be the weighted sum of loss weights for the adversaries $k \in M_t^{(i)}(E_{th})$ and their loss differences from $F(\mathbf{w}^*)$ :*

$$\Omega_t^{(i)}(E_{th}, \delta) = \sum_{k \in M_t^{(i)}(E_{th})} \beta_i^{(k)}(\delta)[F(\mathbf{w}_t(k)) - F(\mathbf{w}^*)]. \tag{7}$$

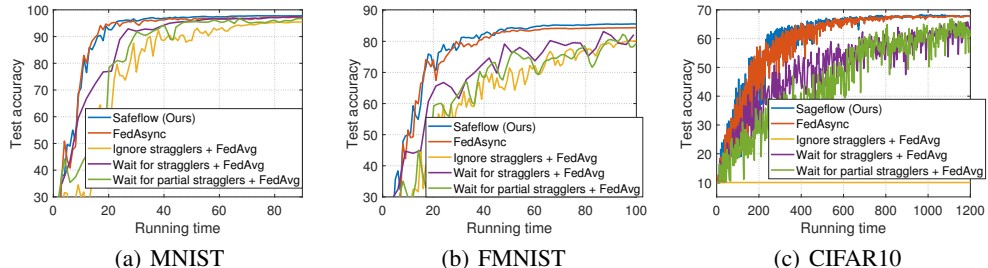

(a) MNIST             (b) FMNIST             (c) CIFAR10

Figure 3: **Performance with only stragglers:** The performance of Sageflow is comparable to FedAsync while outperforming other schemes. However, as to be shown in Fig. 5, FedAsync performs well with only stragglers; it does not integrate naturally with methods that can counter adversaries.

Based on the above assumptions and definition, we state the following theorem which provides the convergence bound of Sageflow in terms of staleness exponent $\lambda$, entropy threshold $E_{th}$ and loss exponent $\delta$. The proof is in Section L of Supplementary Material.

**Theorem 1** *Suppose Assumptions 1, 2 hold and the learning rate $\eta$ is set to be less than $\frac{1}{L}$. Then Sageflow satisfies*

$$\mathbb{E}[F(\mathbf{w}_T) - F(\mathbf{w}^*)] \leq \nu^T[F(\mathbf{w}_0) - F(\mathbf{w}^*)] + (1 - \nu^T)Z(\lambda, E_{th}, \delta) \tag{8}$$

*where $\nu = 1 - \gamma + \gamma(1 - \eta\mu)^E$,*

$$Z(\lambda, E_{th}, \delta) = \frac{\rho_1 + 2\mu G_{max}(\lambda) + 2\mu\Omega_{max}(E_{th}, \delta)}{2\eta\mu^2}, \tag{9}$$

$$G_{max}(\lambda) = \max_{1 \leq t \leq T} \sum_{i=0}^{t-1} \alpha_t^{(i)}(\lambda)e_t^{(i)}, \ \Omega_{max}(E_{th}, \delta) = \max_{t, i \leq t} \Omega_t^{(i)}(E_{th}, \delta), \ e_t^{(i)} = F(\mathbf{w}_i) - F(\mathbf{w}_t).$$

In (8) above, we see a trade-off between $\nu^T$, which determines the convergence speed, and $(1 - \nu^T)Z$, which represents an error. Note that $\nu$ is a function of both $\gamma$, which is the weight on the new model during time-averaging of models in (3), and $E$, which is the number of local updates. If we increase $\gamma$, we have a higher convergence speed (i.e., a small $\nu^T$) but a larger error term, which is the same tradeoff observed in [25] when there are only stragglers. With adversaries also considered, however, our scheme allows a separate control on the error term $Z(\lambda, E_{th}, \delta)$ through staleness exponent $\lambda$, entropy-filtering threshold $E_{th}$ and the loss-weighted exponent $\delta$. Here, in $Z(\lambda, E_{th}, \delta)$ of (9), $G_{max}(\lambda)$ is the error term caused by stragglers controlled by $\lambda$, and $\Omega_{max}(E_{th}, \delta)$ is the error caused by adversaries controlled by $E_{th}$ and $\delta$. First, to gain insights on $G_{max}(\lambda)$, assume that the loss of the global model decreases as global round increases, i.e., $F(\mathbf{w}_i) < F(\mathbf{w}_j)$ for $i > j$. Then, we have $e_t^{(0)} > e_t^{(1)} > ..., > e_t^{(t-1)} > 0$. In order to reduce $\sum_{i=0}^{t-1} \alpha_t^{(i)}(\lambda)e_t^{(i)}$, we have to increase the staleness exponent $\lambda$ to increase $\alpha_t^{(i)}(\lambda)$ for a large $i$ (weight for the group with small staleness) while reducing $\alpha_t^{(i)}(\lambda)$ for a small $i$ (weight for the group with large staleness). By choosing an appropriate $\lambda$, we can control the error term $G_{max}$ while taking advantage of the results sent from stragglers. As for the adversary-induced error $\Omega_{max}(E_{th}, \delta)$, by imposing a threshold $E_{th}$, we can reduce the portion of adversaries (with high entropies) in each group $U_t^{(i)}(E_{th})$, which in turn reduces $\Omega_t^i$ according to (7). Likewise, by introducing $\delta$, we can reduce the loss-weights $\beta_i^{(k)}(\delta)$ (defined in (6)) of the adversaries with large losses, which again reduces $\Omega_t^{(i)}$ according to (7). These effects can significantly reduce $\Omega_{max}$ of (9) compared to the scheme that does not utilize the entropy/loss based idea ($E_{th} = \infty$, $\delta = 0$). A more detailed discussion on the effects of $\lambda$, $E_{th}$, $\delta$ on $G_{max}(\lambda)$ and $\Omega_{max}(E_{th}, \delta)$ is described in Sections M and N of Supplementary Material. In the next section, we show via experiments that Sageflow in fact successfully combats both stragglers and adversaries simultaneously and achieves fast convergence with a small error term.

## 4   Experiments

In this section, we validate Sageflow on MNIST [10], FMNIST [23] and CIFAR10 [8]. The dataset is split into 60,000 train and 10,000 test samples for MNIST and FMNIST, and split into 50,000 train and

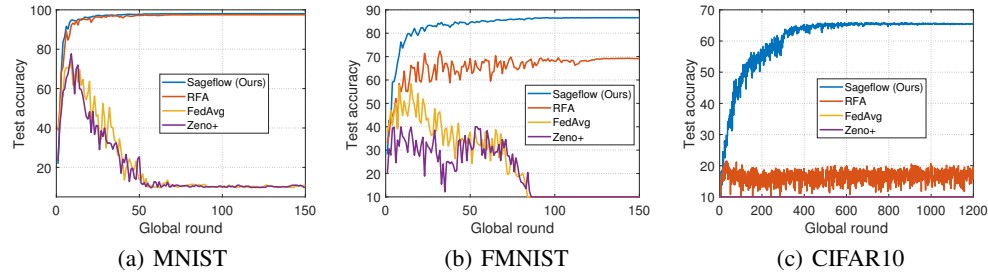

|       (a) MNIST       |       (b) FMNIST       |       (c) CIFAR10       |

Figure 4: **Performance with only adversaries (model poisoning):** Sageflow shows better over performance than other schemes.

Table 1: **Performance with only adversaries**

| | Model poisoning (Test accuracy) | | | Data poisoning (Test accuracy) | | | Scaled backdoor attack (Attack success rate) | | |
|---|---|---|---|---|---|---|---|---|---|
| Methods\Datasets | MNIST | FMNIST | CIFAR10 | MNIST | FMNIST | CIFAR10 | MNIST | FMNIST | CIFAR10 |
| FedAvg | 22.16% | 33.70% | 10.00% | 93.58% | 81.54% | 58.02% | 99.99% | 99.88% | 98.07% |
| Zeno+ | 24.09% | 31.08% | 10.00% | 94.11% | 84.03% | 57.88% | 0.45% | 2.94% | 3.07% |
| RFA | 95.9% | 68.01% | 16.43% | 96.73% | 83.07% | **65.02%** | 67.25% | 66.6% | 86.86% |
| **Sageflow (Ours)** | **97.55%** | **86.65%** | **65.47%** | **97.60%** | **85.14%** | 64.46% | **0.39%** | **1.71%** | **2.14%** |

10,000 test samples for CIFAR10. A simple convolutional neural network (CNN) with 2 convolutional layers and 2 fully connected layers is utilized for MNIST, while CNN with 2 convolutional layers and 1 fully connected layer is used for FMNIST. When training with CIFAR10, we utilized VGG-11.

We consider $N = 100$ devices each having the same number of data samples. We randomly assigned two classes to each device as in [18] to create non-IID situations. We ignored the batch normalization layers when training VGG-11 with CIFAR10. At each global round, we randomly selected a fraction $C$ of devices in the system to participate. For the proposed Sageflow method, we sample $2\%$ of the entire training data uniformly at random to be the public data and performed FL with the remaining $98\%$ of the train set. We note that Sageflow performs well even with a smaller amount of public data (less than 2%) and even with a class-imbalanced public data, as described in Supplementary Material. The number of local epochs at each device is set to 5. The local batch size is set to 10 for all experiments except for the backdoor attack. In addition, we used tuned hyperparameters for Sageflow and other comparison schemes; the details are described in Supplementary Material. Here, we emphasize that the performance of Sageflow is not highly sensitive to the hyperparameters such as loss exponent $\delta$ and entropy threshold $E_{th}$, as long as they are chosen in a reasonable range. These results are also shown in Section D of Supplementary Material.

**Experiments with stragglers.** To confirm the advantage of Sageflow, we first consider the scenario with only the stragglers. The adversaries are not considered here. With only stragglers, we compare Sageflow with the following methods. First is the *wait for stragglers* approach where FedAvg is applied after waiting for all the devices at each global round. The second scheme is the *ignore stragglers* approach where FedAvg is applied after waiting for a certain timeout threshold and ignore the results sent from slow devices. The third scheme is the *wait for a percentage of stragglers* where FedAvg is applied after waiting for a specific portion of the selected devices in each global round. In the main manuscript, we consider a scheme that waits for 50% of selected devices, while the results with other portions are shown in Section K of Supplementary Material. Finally, we consider the asynchronous scheme (FedAsync) [25] where the global model is updated every time the result of each device arrives. For Sageflow and FedAsync, $\gamma$ is decayed while the learning rate is decayed in other schemes.

In Fig. 3, we plot the test accuracy versus running time on different datasets with $C = 0.1$. For a fair comparison, the global aggregation at the server is performed with every $T_d = 1$ periodically for Sageflow and other comparison schemes (ignore stragglers, FedAsync). To model stragglers, each device can have delay of 0, 1, 2 global rounds which is determined independently and uniformly random. In other words, at each global round $t$, we have $S_t = U_t^{(t)} \cup U_{t+1}^{(t)} \cup U_{t+2}^{(t)}$. Experimental results in a more severe straggler scenario (delay of 0 to 8 global rounds) is shown in Section I of Supplementary Material. Our first observation from Fig. 3 is that the *ignore stragglers* scheme can lose significant data at each round and often converges to a suboptimal point with less accuracy. The *wait for stragglers* scheme requires the largest running time until convergence due to the delays caused

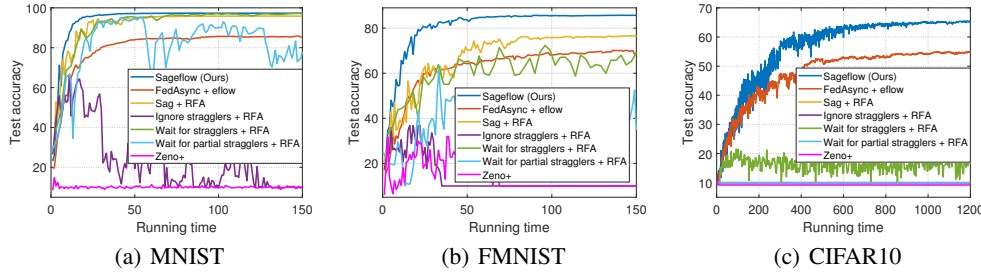

(a) MNIST  (b) FMNIST  (c) CIFAR10

Figure 5: **Performance with both stragglers and adversaries (model poisoning):** Sageflow outperforms various combinations aiming to handle stragglers/adversaries.

Table 2: **Performance with both stragglers and adversaries**

| Methods\Datasets | Model poisoning (Test accuracy) | | | Data poisoning (Test accuracy) | | | Scaled backdoor attack (Attack success rate) | | |
|---|---|---|---|---|---|---|---|---|---|
| | MNIST | FMNIST | CIFAR10 | MNIST | FMNIST | CIFAR10 | MNIST | FMNIST | CIFAR10 |
| Zeno+ | 9.96% | 9.95% | 9.21% | 12.27% | 10.00% | 10.00% | - | - | - |
| Ignore stragglers + RFA | 10.04% | 10.00% | 10.00% | 97.38% | 80.78% | 64.51% | - | - | - |
| Wait for partial stragglers + RFA | 78.29% | 35.29% | 10.00% | 97.59% | 70.15% | 62.13% | 32.16% | 98.49% | 82.26% |
| Wait for stragglers + RFA | 96.20% | 70.01% | 17.17% | 96.60% | 80.01% | 58.51% | 3.17% | 42.79% | 81.74% |
| Sag + RFA | 95.78% | 73.64% | 10.00% | **97.60**% | 83.55% | 64.18% | 99.97% | 99.97% | 90.79% |
| FedAsync + eflow | 84.7% | 66.38% | 53.03% | 83.47% | 48.80% | 51.28% | 99.99% | 99.86% | 91.34% |
| **Sageflow (Ours)** | **97.27**% | **85.23**% | **63.76**% | 97.38% | **85.01**% | **64.87**% | **0.21**% | **3.79**% | **5.56**% |

by slow devices. The *wait for a percentage of stragglers* scheme achieves a similar performance with *wait for stragglers* scheme. Finally, it is observed that the performance of Sageflow is comparable with FedAsync, the state-of-the-art straggler-mitigating scheme. However, FedAsync is not an ideal candidate for our target scenario with both stragglers and adversaries, which will be shown in Fig. 5.

**Experiments with adversaries.** Next, we confirm the performance of Sageflow in Fig. 4 under a synchronous scenario with only adversaries. We compare our method with geometric-median-based RFA [19] and FedAvg under the model update/data poisoning and backdoor attacks. We also consider Zeno+ [27] which utilizes public data at the server: the server first computes the difference between the loss of the received model and the loss of the previous global model. If this loss difference is above a certain threshold, the corresponding model is filtered out. Since there are no stragglers in Fig. 4, we considered a synchronized version of Zeno+ where the models are aggregated via FedAvg after the filtering process. For a fair comparison, we let $2\%$ of the training data to be the public data and the remaining $98\%$ to be distributed at the devices, as in our Sageflow. Comparison with other robust aggregation methods including Multi-Krum is done in Section E of Supplementary Material. For model poisoning, each adversarial device sends $-0.1\mathbf{w}$ to the server, instead of sending the true model $\mathbf{w}$. For data poisoning attack, we conduct *label-flipping* [2], where each label $i$ is flipped to label $i + 1$. For both attacks, we set $C$ to 0.2 and the portion of adversarial devices is assumed to be $r = 0.2$ at each global round. Additional experimental results with a varying portion of adversaries $r$ are reported in Section J of Supplementary Material.

For the backdoor, we use the *model replacement* method (scaled backdoor attack) [1] in which adversaries transmit the scaled version of the corrupted model to replace the global model with a bad model. We conduct *pixel-pattern backdoor attack* [6] in which the specific pixels are embedded in a fraction of images, where these images are classified as a targeted label. See Section B of Supplementary Material for the detailed setting. We measure the attack success rate of the backdoor task by embedding the pixel-pattern into all test samples (except data with label 2) and then comparing the predicted label with the target label. We applied backdoor attack in every global round after the 10-th round for MNIST and FMNIST, and after the 1000-th round for CIFAR10.

In Fig. 4 and Table 1, we compare performance of different schemes with only adversaries. The performance of the table is obtained at a specific global round (see Section A of Supplementary Material for description) by averaging the accuracy over 3 independent trials. FedAvg does not work well on all datasets, and the performance of RFA gets worse as the dataset/neural network model become more complex. Although Zeno+ utilizes public data, it does not perform well under model poisoning attack. This is because the models of adversaries cannot be filtered out based on losses in the beginning of training (see Fig. 2(b)), making Zeno+ difficult to filter out the poisoned

models when the scale factor is 0.1. Zeno+ also has lower performance compared to others in data poisoning, since the loss difference between the global model and the model of adversary is not significant. However, in scaled backdoor attack, we have a large loss difference for Zeno+. Hence, Zeno+ performs well under the scaled backdoor attack. Overall, the results confirm the advantage of Sageflow compared to other methods. The results on no-scaled backdoor attack, where the corrupted model is transmitted without scaling, are shown in Section B of Supplementary Material: to summarize the results, Sageflow can slow down the poisoning of the global model.

**Experiments with both stragglers and adversaries.** Finally in Fig. 5 and Table 2, we consider the setup with both stragglers and adversaries. We compare Sageflow with various straggler/adversary defense combinations, including Zeno+. We consider an asynchronous version of Zeno+ as in [27]: the server first subtracts each survived model (after filtering) from the previous global model to obtain the model difference. Then, the global model is updated asynchronously based on each model difference. Comparison with the Multi-Krum is illustrated in Supplementary Material. We set $C = 0.2$, $r = 0.2$ for model/data poisoning and $C = 0.1$, $r = 0.1$ for the scaled backdoor attack. The stragglers and adversaries are modeled as in Fig. 3 and Table 1, respectively. Note that in scaled backdoor attack, we excluded the results for Zeno+ and RFA combined with *ignore stragglers*, since the models are not trained at all. We have the following observations. First, Zeno+ does not perform well since it does not take both the staleness and entropy into account. It can be also seen that the *wait for stragglers* scheme combined with RFA suffers from the straggler issue. Our next observation is that the RFA combined with *ignore stragglers* method exhibits poor performance. The reason is that the attack ratio could often be very high (larger than $r$) for this deadline-based scheme, which degrades the performance of RFA. Due to the same issue, RFA combined with our staleness-aware grouping (Sag) has performance degradation. FedAsync does not perform well when combined with entropy-based filtering and loss-weighted averaging (eflow), since the model update is conducted one-by-one in the order of arrivals. Due to the same issue, FedAsync cannot be combined with RFA. Overall, the proposed Sageflow performs the best, confirming significant advantages of our scheme under the existence of both stragglers and adversaries.

## 5 Conclusion

We proposed Sageflow, a robust FL scheme that can handle both stragglers and adversaries at the same time. The staleness-aware grouping allows the server to effectively utilize the results sent from stragglers. The grouping-based strategy also integrates naturally with effective defenses against adversary attack. In each grouping stage of our straggler-mitigating approach, entropy-based filtering and loss-weighted averaging function in a highly complementary fashion to protect the system against a wide variety of adversarial attacks. Theoretical convergence analysis provides key insights into why the suggested methods work well. Extensive experimental results show that Sageflow enables robust FL in practical scenarios with a large number of slow devices and adversaries.

An interesting future issue is how to combine existing secure aggregation methods with Sageflow. This presents a challenge as the server should have access to every received individual model to calculate the entropy and loss values in Sageflow.

## Acknowledgments

This work was supported by IITP fund from MSIT of Korea (No. 2020-0-00626) and by National Research Foundation of Korea (No. 2019R1I1A2A02061135).

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
