# Sageflow: Robust Federated Learning against Both Stragglers and Adversaries (Supplementary Material)

**Jungwuk Park**[*]
KAIST
savertm@kaist.ac.kr

**Dong-Jun Han**[*]
KAIST
djhan93@kaist.ac.kr

**Minseok Choi**
Jeju National University
ejaqmf@jejunu.ac.kr

**Jaekyun Moon**
KAIST
jmoon@kaist.edu

## A    Hyperparameter setting

All experiements were performed in PyTorch on an Intel Xeon CPU E5-2620 v4 @ 2.10GHz and a Geforce GTX 1080 Ti.

### A.1    Scenario with only stragglers

The hyperparameter settings for Sageflow are shown in Table 1. For the schemes *ignore stragglers* and *wait for stragglers* combined with FedAvg, we decayed the learning rate during training. For the FedAsync scheme of [7], we take a polynomial strategy with hyperparameters $a = 0.5$, $\alpha = 0.8$, and decayed $\gamma$ during training.

Table 1: Hyperparameters for Sageflow with only stragglers

| Dataset | $\gamma$ | staleness exponent $\lambda$ | loss exponent $\delta$ | entropy threshold $E_{th}$ | Learning rate | $\gamma$ decay |
|---------|----------|------------------------------|------------------------|----------------------------|---------------|----------------|
| MNIST   | 0.5      | 0.5                          | 1                      | 1                          | 0.01          | Every 15 global epochs |
| FMNIST  | 0.5      | 0.5                          | 1                      | 1                          | 0.01          | Every 20 global epochs |
| CIFAR10 | 0.5      | 1.5                          | 1                      | 1                          | 0.01          | Every 300 global epochs |

### A.2    Scenario with only adversaries

**Data poisoning and model poisoning attacks:** Table 2 describes the hyperparameters for Sageflow with only adversaries, under data poisoning and model poisoning attacks. For RFA of [5], the maximum iteration is set to 10. In this setup, the learning rate is decayed for all three schemes (Sageflow, RFA, FedAvg).

**Backdoor attack:** In this backdoor attack scenario, we utilized the Dirichlet distribution with parameter 0.5 for distributing training samples to $N = 100$ devices. The local batch size is set to 64 and the number of poisoned images is 20. The hyperparameter details for Sageflow are shown in Talbe 3.

---

[*]Equal contribution.

35th Conference on Neural Information Processing Systems (NeurIPS 2021).

Table 2: Hyperparameters for Sageflow with only adversaries, under data and model poisoning

| Dataset | $\gamma$ | $\lambda$ | $\delta$ | $E_{th}$ | Learning rate | $\gamma$ decay |
|---------|----------|-----------|----------|----------|---------------|----------------|
| MNIST | 1 | - | 1 | 1 | 0.01 | No decay |
| FMNIST | 1 | - | 1 | 1 | 0.01 | No decay |
| CIFAR10 | 1 | - | 1 | 1 | 0.01 | No decay |

Table 3: Hyperparameters for Sageflow with only adversaries, under backdoor attack

| Dataset | $\gamma$ | $\lambda$ | $\delta$ | $E_{th}$ | Learning rate | $\gamma$ decay |
|---------|----------|-----------|----------|----------|---------------|----------------|
| MNIST | 1 | - | 5 | 2 | 0.01 | No decay |
| FMNIST | 1 | - | 5 | 2 | 0.01 | No decay |
| CIFAR10 | 1 | - | 5 | 2 | 0.01 | No decay |

### A.3 Scenario with both stragglers and adversaries

**Data poisoning and model poisoning attacks:** The hyperparameters for Sageflow are shown in Table 2.

**Backdoor attack:** The hyperparameter details are shown in Table 4.

Table 4: Hyperparameters for Sageflow with both stragglers and adversaries, under backdoor attack

| Dataset | $\gamma$ | $\lambda$ | $\delta$ | $E_{th}$ | Learning rate | $\gamma$ decay |
|---------|----------|-----------|----------|----------|---------------|----------------|
| MNIST | 0.5 | 0.5 | 5 | 2 | 0.01 | No decay |
| FMNIST | 0.5 | 0.5 | 5 | 2 | 0.01 | No decay |
| CIFAR10 | 0.5 | 1.5 | 5 | 2 | 0.01 | Every 1000 global epochs |

### A.4 Setup for Tables 1 and 2 in the main manuscript

In Tables 1 and 2 of the main manuscript, we evaluated test accuracies at specific global rounds or time. We specify these values in Table 5.

## B Additional experiments under backdoor attack

### B.1 Experimental setup with backdoor attack

We embedded 12 white pixels in the top-left corner of the image and the labels of these poisoned images are set to 2. We utilize the Dirichlet distribution with parameter 0.5 for distributing training samples to $N = 100$ devices. We set $C$, the fraction of $N = 100$ devices participating in each global round, to 0.1 and $r$, the portion of adversarial devices, also to 0.1 at each global round. The local batch size is set to 64. The number of poisoned images in a batch is 20, and we do not decay the learning rate here.

### B.2 Experiments under no-scaled backdoor attack

In addition to the *model replacement* backdoor attack (or scaled backdoor attack) we considered so far, we perform additional experiments under the no-scaled backdoor attack [1] where the adversarial devices do not scale the weights and only transmit corrupted models to the server. Fig. 1 shows the performance under the no-scaled backdoor attack with only adversaries (no stragglers). Fig. 2 shows the case with both stragglers and adversaries. We set $C = 0.1$ and $r = 0.1$ for both figures. Since the models corrupted by no-scaled backdoor attack do not significantly degrade the overall test accuracy, it seems that Sageflow and other schemes are not able to completely defend against the attack. However, Sageflow noticeably slows down the poisoning of the global model compared to other methods.

Table 5: Global round or time for evaluating accuracy in Tables 1 and 2 of the main manuscript.

| | Model poisoning | | | Data poisoning | | | Scaled backdoor attack | | |
|---|---|---|---|---|---|---|---|---|---|
| Setup \Datasets | MNIST | FMNIST | CIFAR10 | MNIST | FMNIST | CIFAR10 | MNIST | FMNIST | CIFAR10 |
| Only adversaries (global round) | 40 | 70 | 800 | 40 | 70 | 800 | 180 | 100 | 1600 |
| Both stragglers/adversaries (time) | 40 | 70 | 828 | 70 | 70 | 800 | 50 | 50 | 1200 |

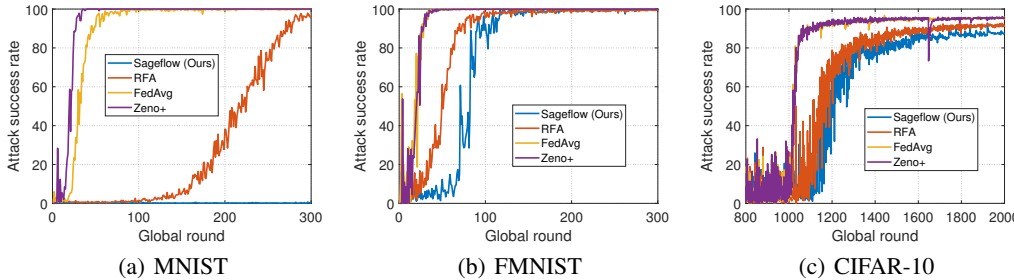

(a) MNIST  (b) FMNIST  (c) CIFAR-10

Figure 1: Performance comparison with only adversaries under no-scaled backdoor attack. Sageflow can slow down the poisoning of the global model compared to other methods.

## C  Experiments under model poisoning with scale 10

Some additional experiments were conducted under model poisoning with the scale factor 10. Fig. 3 shows the results with $C = 0.2$ and with $r = 0.2$. When modeling stragglers, each device can have a delay of 0, 1 or 2, which is determined independently and uniformly random. The loss associated with a poisoned device increases if we increase the scale factor from 0.1 to 10. Hence, not only Sageflow but also Zeno+ can effectively defend against the attacks with only adversaries. However, under the existence of both stragglers/adversaries, Sageflow outperforms other baselines.

## D  Experimental results with varying hyperparameters

To observe the impact of hyperparameter setting, we performed additional experiments with various $\delta$ and $E_{th}$ values, the key hyperparameters of Sageflow. The results are shown in Fig. 4 with only adversaries. We performed the data poisoning attack for varying $\delta$ and the model poisoning attack with scale 0.1 for varying $E_{th}$. We set $C = 0.2$ and $r = 0.2$.

First, the results under data poisoning show that the performance of Sageflow is not sensitive to $\delta$ if they are chosen in the appropriate range of $[1, 2]$. For the model poisoning attack, if we use a very small $E_{th}$ like 0.3, the performance is poor because a large number of devices get filtered out. If we use a large $E_{th}$, the performance is also very poor since the scheme cannot filter out the adversaries. However, similar to the behavior of hyperparameter $\delta$, we can confirm that our scheme performs well regardless of dataset if $E_{th}$ is chosen in an appropriate range of $[1, 2]$.

To summarize, our scheme still performs well (better than RFA), even with coarsely chosen hyperparameter values regardless of the dataset.

## E  Comparison with other robust aggregation methods against adversaries

In this section, we compare our algorithm with various existing aggregation methods that are robust against adversaries.

### E.1  Performance comparison with Multi-Krum

While we compared Sageflow with RFA in our main manuscript, here we compare our scheme with *Multi-Krum* [2] which is a Byzantine-resilient aggregation method targeting conventional distributed learning setup with IID data across nodes. In Multi-Krum, among $N$ workers in the system, the server tolerates $f$ Byzantine workers under the assumption of $2f + 2 < N$. After filtering $f$ worker nodes

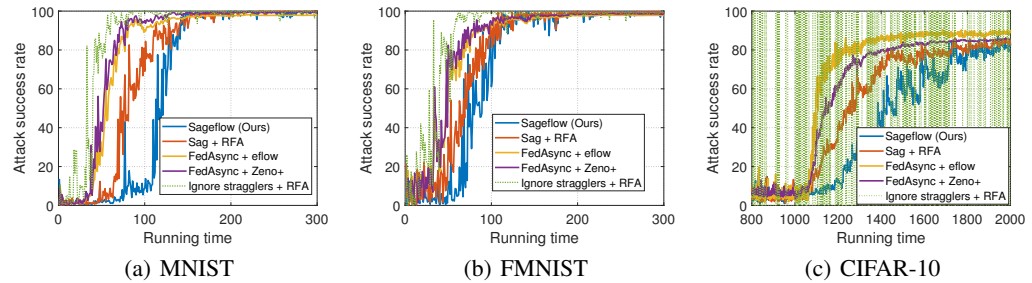

Figure 2: Performance with both stragglers and adversaries under no-scaled backdoor attack. Sageflow can slow down the poisoning of the global model compared to other methods.

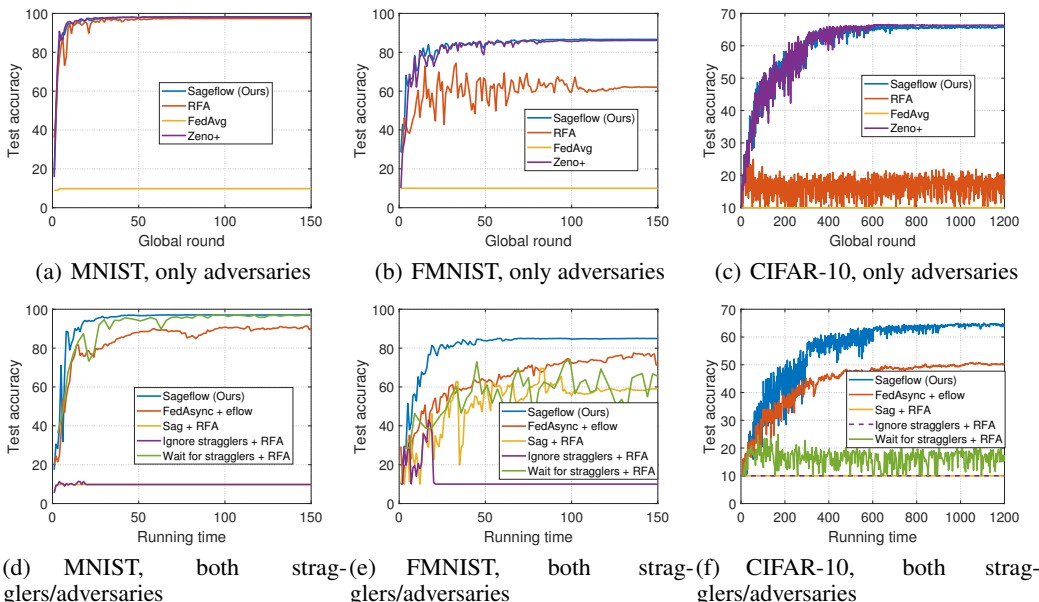

Figure 3: Performance under model poisoning with scale 10. Both Sageflow and Zeno+ perform well with only adversaries, while only Sageflow performs well under the existence of both stragglers/adversaries.

based on squared-distances, the server chooses $M$ workers among $N - f$ remaining workers with the best scores and aggregates them. We set $M = N - f$ for comparing our scheme with Multi-Krum.

Fig. 5 compares Sageflow with Multi-Krum under model poisoning with scale 10. The stragglers are modeled with delay 0, 1, 2. We first observe Figs. 5(a) and 5(b) which show the results with only adversaries. It can be seen that if the number of adversaries exceed $f$, the performance of Multi-Krum drops dramatically. Compared to Multi-Krum, the proposed Sageflow method can filter out the poisoned devices and then take the weighted sum of the survived results even when the portion of adversaries is high. Figs. 5(c) and 5(d) show the results under the existence of both stragglers and adversaries, under the model poisoning attack. We let $C = 0.2$ and $r = 0.2$, and the parameter $f$ of Multi-Krum is set to the maximum value satisfying $2f + 2 < N$, where $N$ depends on the number of received results for both staleness-aware grouping (Sag) and *ignore stragglers* approaches. However, even when we set $f$ to the maximum value, the number of adversaries can still exceed $f$, which degrades the performance of Multi-Krum combined with staleness-aware grouping (Sag) or the *ignore stragglers* approach. Obviously, Multi-Krum can be combined with the *wait for stragglers* strategy by setting $f$ large enough. However, this scheme still suffers from the effect of stragglers, which significantly slows down the overall training process.

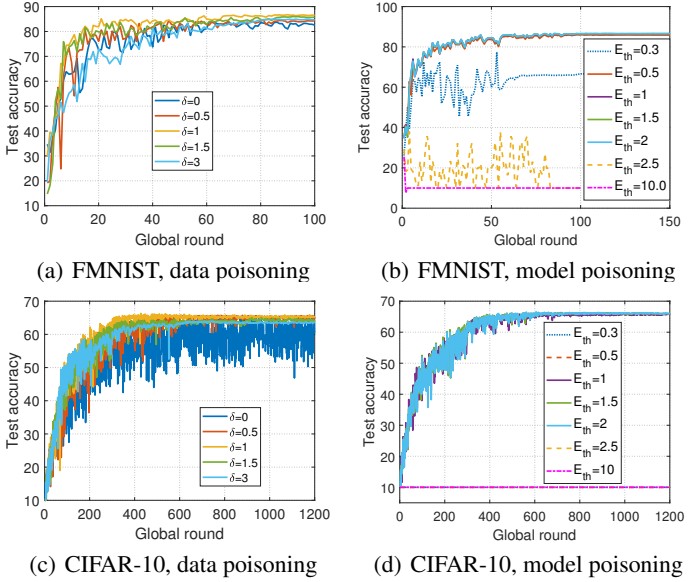

(a) FMNIST, data poisoning      (b) FMNIST, model poisoning

(c) CIFAR-10, data poisoning      (d) CIFAR-10, model poisoning

Figure 4: Impact of varying hyperparameter values under model poisoning and data poisoning attacks. The performance of Sageflow is not highly sensitive to the exact settings of loss exponent $\delta$ and entropy threshold $E_{th}$, as long as they are chosen in a reasonable range.

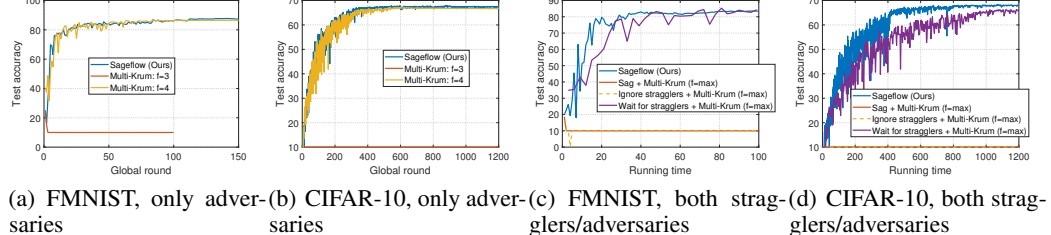

(a) FMNIST, only adversaries    (b) CIFAR-10, only adversaries    (c) FMNIST, both stragglers/adversaries    (d) CIFAR-10, both stragglers/adversaries

Figure 5: Performance comparison with Multi-Krum under model poisoning. With only adversaries, Multi-Krum performs well when an appropriate $f$ parameter value is chosen. However, the performance of Multi-Krum degrades significantly when stragglers exist (even when combined with straggler-mitigating schemes). This is because the attack ratio can become very high when combined with staleness-aware grouping or the ignoring stragglers scheme; the number of adversaries exceeds $f$, significantly degrading the performance of Multi-Krum. When Multi-Krum is combined with the wait for stragglers scheme, the performance is not degraded by adversaries but by waiting for slow devices.

Fig. 6 compares Sageflow with Multi-Krum under scaled backdoor attack. The portion of participating devices and the portion of adversaries at each global round are $C = 0.1$ and $r = 0.1$, respectively. The results are consistent with the results in Fig. 5, confirming the advantage of Sageflow over Multi-Krum combined with straggler-mitigating schemes.

## E.2 Performance comparison with FLTrust

We performed additional experiments by comparing our scheme with FLTrust proposed in [4] and OracleSGD in [6]. FLTrust utilizes public data at the server to update the server model and compute cosine similarities between each local model and the server model. By aggregating local models based on the cosine similarity score, FLTrust enables to reduce the impact of adversaries. OracleSGD can be viewed as an ideal performance as it assumes the full knowledge on which devices are the adversaries.

The setup is exactly the same in Fig. 5 of our main manuscript. We considered model poisoning attack with scale 0.1. To this end, we applied FLtrust and OracleSGD in each grouping stage of our

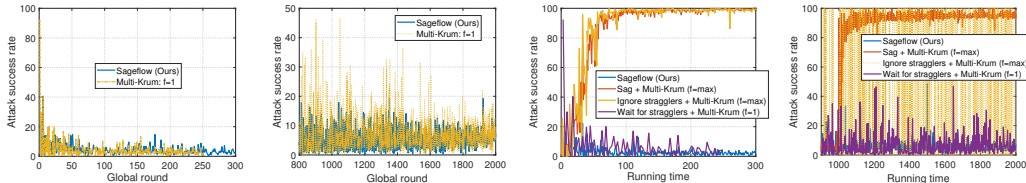

(a) FMNIST, only adver-saries  (b) CIFAR-10, only adver-saries  (c) FMNIST, both strag-glers/adversaries  (d) CIFAR-10, both strag-glers/adversaries

Figure 6: Performance comparison with Multi-Krum under scaled backdoor attack. Multi-Krum performs well with only adversaries, but the performance is degraded when combined with straggler-mitigating schemes under the existence of both stragglers and adversaries.

staleness-aware aggregation. Since FLTrust also utilizes public data, we allocated the same public data for FLTrust as Sageflow. For OracleSGD, we performed FedAvg using the models of the benign devices. When public data is class-balanced (the number of samples are distributed uniformly across the classes in the public data), our scheme achieves the accuracy of 86.54% at running time 120 on FMNIST while 86.56%, 86.91% are achievable for FLTrust, OracleSGD, respectively. However, under the setting of Fig. 9 in Supplementary Material (when the public data is class-imbalanced), our scheme achieves accuracy of 85.6% at running time 120 on FMNIST while 81.5%, 86.91% are achievable for FLTrust, OracleSGD, respectively. Now using CIFAR10 with a class-balanced public data, our scheme and OracleSGD achieve 66.48% and 66.05% respectively, while FLTrust does not work well (achieving accuracy of 10%) on this relatively complicated dataset under our severe non-IIDness scenario. The overall results confirm the advantage of Sageflow in various practical settings.

### E.3   Performance comparison with DiverseFL

We also performed additional experiments using DiverseFL proposed in [6]. In DiverseFL, the server utilizes each of the received "local dataset" to compute the gradient of each client. Then, the server computes the similarity between each of the computed gradient and the received gradient sent from the corresponding client, to filter out adversaries. We utilized DiverseFL in each grouping stage of our staleness-aware grouping to compare with our Sageflow. The setup is the same setup as in Fig. 5 with both stragglers and adversaries. For a fair comparison, we let each client to send 2% of its local dataset to the server in DiverseFL. For MNIST, our scheme achieves accuracy of 97.71%, while DiverseFL achieves 97.58%. For FMNIST, the accuracies are 86.54% and 85.55% for our scheme and DiverseFL, respectively. Finally, 87.89% and 85.94% are achieved for our scheme and DiverseFL using FEMNIST dataset.

Here we note that DiverseFL requires several additional constraints to be utilized in practice compared to our Sageflow. First of all, in order for the server to compute the similarity between gradients, it is essential for the clients participating in FL to directly send their local data to the server. Secondly, the adversaries may upload the corrupted data to the server, which makes DiverseFL challenging to combat adversaries. Although the authors of this paper claim that the group of experts can remove this corrupted data at the server, this requires additional resources and efforts. Moreover, in order for the server to compute the gradient of the clients, DiverseFL requires the server to distinguish the local datasets of all clients. In other words, the server should remember which dataset come from which client, which can be challenging in cross-device FL scenarios having a significant number of clients in the system. Finally, at the server-side, DiverseFL needs to perform a large number of forward/backward propagations for computing the gradient (after multiple updates) of all clients in each global round, which causes additional computational burden at the server.

## F   Experimental results on the effect of loss-weighted averaging and entropy-based filtering

In Fig. 7, we observe the effect of loss-weighted averaging and entropy-based filtering with only adversaries. For the model poisoning, we performed attack with scale 0.1. We also let $C = 0.2$ and

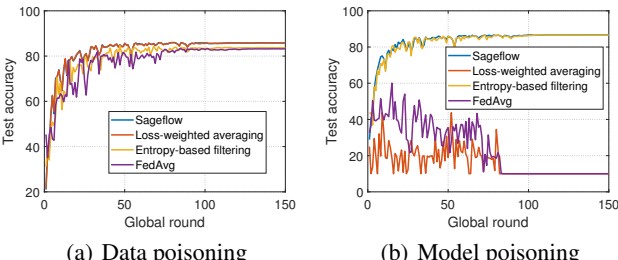

(a) Data poisoning        (b) Model poisoning

Figure 7: Effect of loss-weighted averaging and entropy-based filtering with only adversaries. FMNIST dataset used. Both schemes work in a highly complementary fashion to tackle various attacks. Utilizing only one of these methods significantly degrade the model performance.

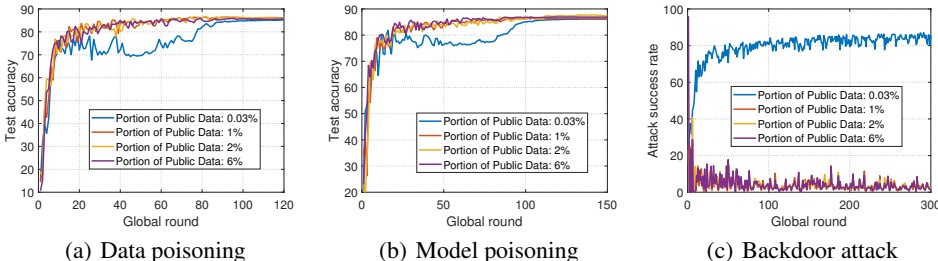

(a) Data poisoning        (b) Model poisoning        (c) Backdoor attack

Figure 8: Impact of varying portions of public data at the server using FMNIST. We set $C = 0.1$, $r = 0.1$ for the backdoor attack and $C = 0.2$, $r = 0.2$ for other cases. Sageflow can defend against various attacks with only a small portion of public data (1%).

$r = 0.2$. Both schemes work in a highly complementary fashion to tackle various attacks. Utilizing only one of these methods significantly degrades the model performance.

## G  Impact of public data

### G.1  Varying amount of public data

Fig. 8 shows the results with various portions of public data on FMNIST. We set $C = 0.1$, $r = 0.1$ for the backdoor attack and $C = 0.2$, $r = 0.1$ for others. Note that in the main manuscript, we let 2% of the entire training set to be the public data and then the remaining data to be the training data at the devices for fair comparison with other schemes. In the setting of Fig. 8, the whole training set is utilized at the devices for federated learning, and each device sends a certain portion of its local data (for example, 2%) to the server to construct public dataset. It can be seen that our Sageflow protects the system against adversarial attacks using only a very small amount of public data. When the portion of the public data used gets as small as 0.03% of the entire training set, the robustness of Sageflow does seem to suffer, but at 1% and higher, the performance is very robust across the board.

### G.2  Imbalanced public data

In Fig. 9, we observe the performance of Sageflow with imbalanced public data; the number of data samples are different across the classes in the public data. We utilize the Dirichlet distribution with parameter 0.5 and 3 for distributing training samples to $N = 100$ devices. We set $C = 0.2$ and $r = 0.2$. We also let 2% of the entire training set to the public data. The overall results show that Sageflow performs better than other schemes even with imbalanced public data at the server.

## H  Experiments on other datasets

### H.1  Experiments on FEMNIST

We performed additional experiments using FEMNIST dataset [3] and obtained consistent results: under the same setting of Fig. 5 in the main manuscript with both stragglers and adversaries (model

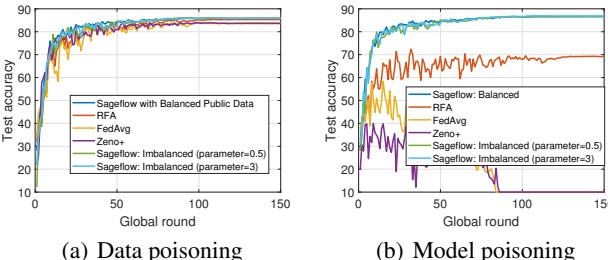

(a) Data poisoning         (b) Model poisoning

Figure 9: Impact of imbalanced public data at the server using FMNIST. We set $C = 0.2$, $r = 0.2$. We let $2\%$ of the entire dataset to be the public data. Sageflow performs better than other schemes even with an imbalanced public data.

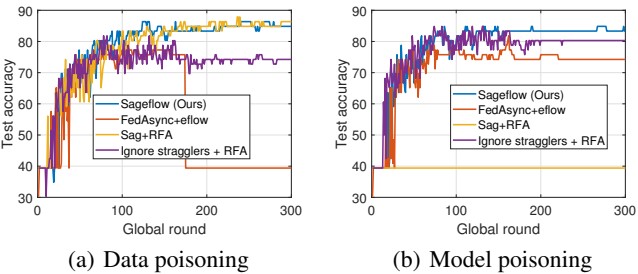

(a) Data poisoning         (b) Model poisoning

Figure 10: Performance of different schemes on a medical dataset (Covid-19 image dataset) under data and model poisoning attacks. Only Sageflow can handle both types of attacks in the presence of stragglers as well.

poisoning), our scheme achieves accuracy of 86.78% at running time 100 while 80.19%, 76.17%, 10%, 10% are achievable for FedAsync + eflow, Sag + RFA, ignore stragglers + RFA, wait for stragglers + RFA, respectively. These results further confirm the advantage of our scheme compared to various baselines.

### H.2 Experiments on Covid-19 dataset open to public

We performed additional experiments on Kaggle's Covid-19 dataset[2], which is *open to public*. Image classification is performed to detect Covid-19 using Chest X-ray images. The dataset consists of 317 color images of $3480 \times 4248$ pixels in 3 classes (Normal, Covid and Viral-Pneumonia). There are a total of 251 training images and 66 test images. We took 15 image samples to construct the server data, 5 samples for each class. Given only 251 training samples, this corresponds to a 6% of the entire training set. We wanted to go to a lower portion, but 5 samples per class was as low as one could reasonably go for estimating model entropies and losses at the server. We divided the remaining training samples into 10 distributed devices, so each device got 23 or 24 image samples over 3 classes. This setup simulates a realistic scenario, where a number of individual patients or private clinics, each having some example X-ray images, wish to collaborate in developing a learner that would classify new images. In the process, the server (e.g., at a central hospital or a service provider) utilizes anonymous public X-ray image samples of the same disease categories to provide protection against adversary attacks.

We set the participating device portion to $C = 1$ and the adversary portion to $r = 0.1$. We assumed both model poisoning and data poisoning attacks. For the model poisoning, we used scale 0.1. We also allowed stragglers in the system: each device could have a delay of 0 or 1, as determined independently and uniformly random. We resized the images into $224 \times 224$ pixels and employed convolutional neural networks with 6 convolutional layers and 1 fully connected layer.

Fig. 10 shows the results. It is clear that only Sageflow can combat both types of attacks effectively under the existence of stragglers as well.

---

[2]https://www.kaggle.com/pranavraikokte/covid19-image-dataset

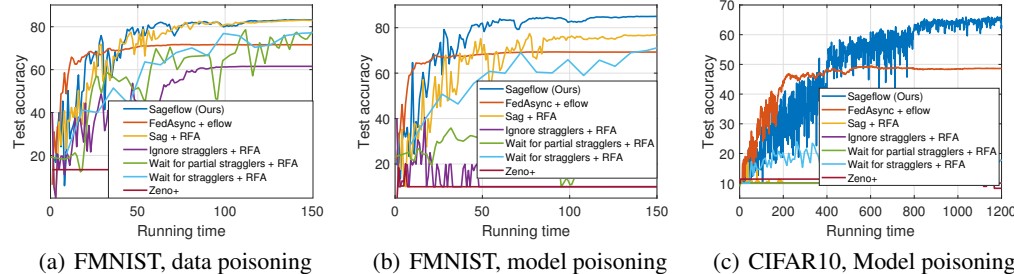

(a) FMNIST, data poisoning     (b) FMNIST, model poisoning     (c) CIFAR10, Model poisoning

Figure 11: Performance in a more severe straggler scenario where each device can have delay of $0$ to $8$. We set $C = 0.4$, $r = 0.2$. Sageflow still performs better than other schemes in a severe straggler scenario.

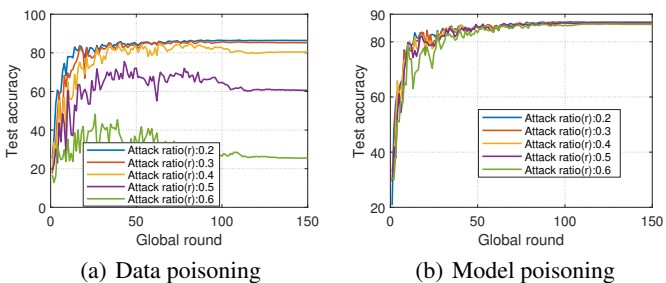

(a) Data poisoning            (b) Model poisoning

Figure 12: Performance with varying portions of adversaries. Data and model poisoning attacks are considered with FMNIST. We set $C = 0.2$.

# I   Experiments in a more severe straggler scenario

When modeling stragglers, we gave a delay of $0$, $1$, $2$ to each device in the experiments of the main manuscript. In this section, each device can have delay of $0$ to $8$, again determined independently and uniformly random. In Fig. 11, we show the results with both stragglers and adversaries under data and model poisoning. We set $C$ to 0.4 and $r$ to 0.2. It can be seen that our Sageflow still shows the best performance under both data poisoning and model poisoning compared to other baseline schemes.

# J   Experiments with varying portion of adversaries

In this section, we show the performance of Sageflow with varying portions of adversaries under data and model poisoning attacks with scale 10. We do not consider stragglers here. We set $\delta$ to 1 and $E_{th}$ to 1 as in the experiments of the main manuscript. Fig. 12 shows the results with different attack ratios on FMNIST. For data poisoning, our Sageflow shows robustness against attack ratios up to 0.4, but with 0.5 or higher, performance is degraded. For model poisoning, it can be seen that our Sageflow performs well even with higher attack ratios.

# K   Additional comparision with waiting for partial stragglers

In Fig. 13, we provide new experimental results by considering a scheme that waits for 30%, 50%, 70% of the selected devices. Here, when the waiting percentage is small so that the time required for one global round is less than our time threshold, we are ignoring more stragglers (which means that the attack ratio becomes higher under the existence of adversaries) so the performance is visibly poor when combined with RFA. As the waiting percentage becomes larger, the method naturally reduces to the wait for stragglers scheme. Overall, new results again confirm significant advantages of our method.

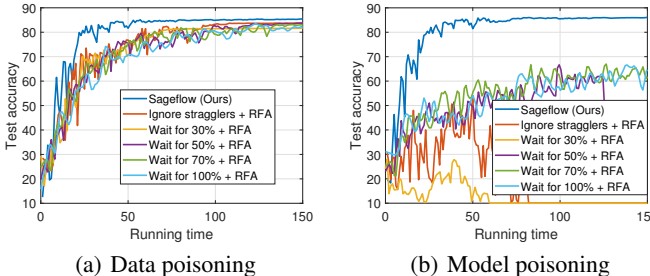

(a) Data poisoning         (b) Model poisoning

Figure 13: Comparison with the scheme that waits for a portion of devices. FMNIST is utilized with $C = 0.2$ and $r = 0.2$.

# L    Proof of Theorem 1

## L.1    Additional Notations for Proof

Let $\mathbf{w}_t^j(k)$ be the model of the $k$-th benign device after $j$ local updates starting from global round $t$. At global round $t$, each device receives the current global model $\mathbf{w}_t$ and round index (time stamp) $t$ from the server, and sets its initial model to $\mathbf{w}_t$, i.e., $\mathbf{w}_t^0(k) \leftarrow \mathbf{w}_t$ for all $k = 1, \ldots, N$. Then each $k$-th benign device performs $E$ local updates of stochastic gradient descent (SGD) with learning rate $\eta$:

$$\mathbf{w}_t^j(k) \leftarrow \mathbf{w}_t^{j-1}(k) - \eta \nabla F_k(\mathbf{w}_t^{j-1}(k), \xi_t^{j-1}(k)) \; for \; j = 1, \ldots, E, \tag{1}$$

where $\xi_t^j(k)$ is a set of data samples that are randomly selected from the $k$-th device during the $j$-th local update at global round $t$. After $E$ local updates, the $k$-th benign device transmits $\mathbf{w}_t^E(k)$ to the server. However, in each round, the adversarial devices transmit poisoned model parameters.

Using these notations, the parameters defined in Section 2 can be rewritten as follows:

$$\mathbf{v}_{t+1}^{(i)} = \sum_{k \in U_t^{(i)}(E_{th})} \beta_i^{(k)}(\delta) \mathbf{w}_i^E(k) \text{ where } \beta_i^{(k)}(\delta) \propto \frac{m_k}{\{F_{pub}(\mathbf{w}_i^E(k))\}^\delta} \text{ and } \sum_{k \in U_t^{(i)}(E_{th})} \beta_i^{(k)}(\delta) = 1 \tag{2}$$

$$\mathbf{z}_{t+1} = \sum_{i=0}^t \alpha_t^{(i)}(\lambda) \mathbf{v}_{t+1}^{(i)} \quad \text{where } \alpha_t^{(i)}(\lambda) \propto \frac{\sum_{k \in U_t^{(i)}} m_k}{(t - i + 1)^\lambda} \text{ and } \sum_{i=0}^t \alpha_t^{(i)}(\lambda) = 1 \tag{3}$$

$$\mathbf{w}_{t+1} = (1 - \gamma)\mathbf{w}_t + \gamma \mathbf{z}_{t+1} \tag{4}$$

We also define

$$\Gamma_t^{(i)} = \sum_{B_t^{(i)}(E_{th})} \beta_i^{(k)}(\delta) \tag{5}$$

where $0 \leq \Gamma_t^{(i)} \leq 1$.

## L.2    Key Lemma and Proof

We introduce the following key lemma for proving Theorem 1. A part of our proof is based on the convergence proof of FedAsync in [7].

**Lemma 1** *Suppose Assumptions 1, 2 hold and the learning rate $\eta$ is set to be less than $\frac{1}{L}$. Consider the $k$-th benign device that received the current global model $\mathbf{w}_t$ from the server at global round $t$. After $E$ local updates, the following holds:*

$$\mathbb{E}[F(\mathbf{w}_t^E(k)) - F(\mathbf{w}^*)|\mathbf{w}_t^0(k)] \leq (1 - \eta\mu)^E[F(\mathbf{w}_t) - F(\mathbf{w}^*)] + \frac{E\rho_1\eta}{2}. \tag{6}$$

*Proof of Lemma 1.* First, consider one step of SGD in the $k$-th local device. For a given $\mathbf{w}_t^j(k)$, for all global round $t$ and for all local updates $j \in \{0, 1, \ldots, E - 1\}$, we have

$$\mathbb{E}[F(\mathbf{w}_t^{j+1}(k)) - F(\mathbf{w}^*)|\mathbf{w}_t^j(k)]$$

$$\leq F(\mathbf{w}_t^j(k)) - F(\mathbf{w}^*) - \eta\mathbb{E}[\nabla F(\mathbf{w}_t^j(k))^T \nabla F_k(\mathbf{w}_t^j(k), \xi_t^j(k))|\mathbf{w}_t^j(k)]$$

$$+ \frac{L\eta^2}{2}\mathbb{E}[\|\nabla F_k(\mathbf{w}_t^j(k), \xi_t^j)\|^2|\mathbf{w}_t^j(k)] \qquad \blacktriangleright \text{ SGD update and } L\text{-smoothness}$$

$$\leq F(\mathbf{w}_t^j(k)) - F(\mathbf{w}^*) + \frac{\eta}{2}\mathbb{E}[\|\nabla F(\mathbf{w}_t^j(k)) - \nabla F_k(\mathbf{w}_t^j(k), \xi_t^j(k))\|^2|\mathbf{w}_t^j(k)]$$

$$- \frac{\eta}{2}\|\nabla F(\mathbf{w}_t^j(k))\|^2 \qquad\qquad\qquad\qquad \blacktriangleright \eta < \frac{1}{L}$$

$$\leq F(\mathbf{w}_t^j(k)) - F(\mathbf{w}^*) - \frac{\eta}{2}\|\nabla F(\mathbf{w}_t^j(k))\|^2 + \frac{\eta\rho_1}{2} \qquad \blacktriangleright \text{ Assumption 2}$$

$$\leq (1 - \eta\mu)[F(\mathbf{w}_t^j(k)) - F(\mathbf{w}^*)] + \frac{\eta\rho_1}{2} \qquad \blacktriangleright \mu\text{-strongly convexity} \quad (7)$$

Applying above result to $E$ local updates in $k$-th local device, we have

$$\mathbb{E}\left[F(\mathbf{w}_t^E(k)) - F(\mathbf{w}^*)|\mathbf{w}_t^0(k)\right]$$

$$= \mathbb{E}[\,\mathbb{E}[F(\mathbf{w}_t^E(k)) - F(\mathbf{w}^*)|\mathbf{w}_t^{E-1}(k)]|\mathbf{w}_t^0(k)\,] \qquad \blacktriangleright \text{ Law of total expectation}$$

$$\leq (1 - \eta\mu)\mathbb{E}[[F(\mathbf{w}_t^{E-1}(k)) - F(\mathbf{w}^*)]|\mathbf{w}_t^0(k)] + \frac{\eta\rho_1}{2} \qquad \blacktriangleright \text{ Inequality (7)}$$

$$\vdots$$

$$\leq (1 - \eta\mu)^E[F(\mathbf{w}_t^0(k)) - F(\mathbf{w}^*)] + \frac{\eta\rho_1}{2}\sum_{j=1}^E (1 - \eta\mu)^{j-1}$$

$$= (1 - \eta\mu)^E[F(\mathbf{w}_t^0(k)) - F(\mathbf{w}^*)] + \frac{\eta\rho_1}{2}\frac{1 - (1 - \eta\mu)^E}{\eta\mu} \qquad \blacktriangleright \text{ From } \eta < \frac{1}{L} \leq \frac{1}{\mu}, \; \eta\mu < 1$$

$$\leq (1 - \eta\mu)^E[F(\mathbf{w}_t) - F(\mathbf{w}^*)] + \frac{E\eta\rho_1}{2} \qquad \blacktriangleright \text{ From } \eta\mu < 1, \; 1 - (1 - \eta\mu)^E \leq E\eta\mu$$

## L.3 Proof of Theorem 1

Now utilizing Lemma 1, we provide the proof for Theorem 1. First, consider one round of global aggregation at the server. For a given $\mathbf{w}_{t-1}$, the server updates the global model according to equation

(4). Then for all $t \in 1, \ldots, T$, we have

$$\mathbb{E}[F(\mathbf{w}_t) - F(\mathbf{w}^*)|\mathbf{w}_{t-1}]$$

$$\overset{(a)}{\leq} (1-\gamma)[F(\mathbf{w}_{t-1}) - F(\mathbf{w}^*)] + \gamma\mathbb{E}[F(\mathbf{z}_t) - F(\mathbf{w}^*)|\mathbf{w}_{t-1}]$$

$$\overset{(b)}{\leq} (1-\gamma)[F(\mathbf{w}_{t-1}) - F(\mathbf{w}^*)] + \gamma\sum_{i=0}^{t-1}\alpha_{t-1}^{(i)}(\lambda)\mathbb{E}[F(\mathbf{v}_t^{(i)}) - F(\mathbf{w}^*)|\mathbf{w}_{t-1}]$$

$$\overset{(c)}{\leq} (1-\gamma)[F(\mathbf{w}_{t-1}) - F(\mathbf{w}^*)] + \gamma\sum_{i=0}^{t-1}\alpha_{t-1}^{(i)}(\lambda)\sum_{k\in U_{t-1}^{(i)}(E_{th})}\beta_i^{(k)}(\delta)\mathbb{E}[F(\mathbf{w}_i^E(k)) - F(\mathbf{w}^*)|\mathbf{w}_{t-1}]$$

$$= (1-\gamma)[F(\mathbf{w}_{t-1}) - F(\mathbf{w}^*)] + \gamma\sum_{i=0}^{t-1}\alpha_{t-1}^{(i)}(\lambda)\bigg\{\sum_{k\in B_{t-1}^{(i)}(E_{th})}\beta_i^{(k)}(\delta)\mathbb{E}[F(\mathbf{w}_i^E(k)) - F(\mathbf{w}^*)|\mathbf{w}_{t-1}]$$

$$+ \sum_{k\in M_{t-1}^{(i)}(E_{th})}\beta_i^{(k)}(\delta)\mathbb{E}[F(\mathbf{w}_i^E(k)) - F(\mathbf{w}^*)|\mathbf{w}_{t-1}]\bigg\}$$

$$\overset{(d)}{\leq} (1-\gamma)[F(\mathbf{w}_{t-1}) - F(\mathbf{w}^*)] + \frac{E\eta\rho_1\gamma}{2} + \gamma\Omega_{max}(E_{th}, \delta)$$

$$+ \gamma(1-\eta\mu)^E\sum_{i=0}^{t-1}\alpha_{t-1}^{(i)}(\lambda)\sum_{k\in B_{t-1}^{(i)}(E_{th})}\beta_i^{(k)}(\delta)\left[\underbrace{F(\mathbf{w}_i) - F(\mathbf{w}^*)}_{F(\mathbf{w}_i)-F(\mathbf{w}_{t-1})+F(\mathbf{w}_{t-1})-F(\mathbf{w}^*)}\right]$$

$$= (1-\gamma+\gamma\sum_{i=0}^{t-1}\alpha_{t-1}^{(i)}(\lambda)\Gamma_{t-1}^{(i)}(1-\eta\mu)^E)[F(\mathbf{w}_{t-1}) - F(\mathbf{w}^*)] + \frac{E\eta\rho_1\gamma}{2} + \gamma\Omega_{max}(E_{th}, \delta)$$

$$+ \gamma(1-\eta\mu)^E\sum_{i=0}^{t-2}\alpha_{t-1}^{(i)}(\lambda)\sum_{k\in B_{t-1}^{(i)}(E_{th})}\beta_i^{(k)}(\delta)\left[F(\mathbf{w}_i) - F(\mathbf{w}_{t-1})\right]$$

$$\overset{(e)}{\leq} (1-\gamma+\gamma(1-\eta\mu)^E)[F(\mathbf{w}_{t-1}) - F(\mathbf{w}^*)] + \frac{E\eta\rho_1\gamma}{2} + \gamma\Omega_{max}(E_{th}, \delta) + \gamma G_{t-1}(\lambda) \qquad (8)$$

where $e_t^{(i)} := F(\mathbf{w}_i) - F(\mathbf{w}_t)$ and $G_t(\lambda) := \sum_{i=0}^{t-1}\alpha_t^{(i)}(\lambda)e_t^{(i)}$ and $G_0(\lambda) = 0$. $(a)$, $(b)$, $(c)$ come from convexity, $(d)$ follows Lemma 1 and the definition $\Omega_{max} = \max_{0\leq i\leq t, 0\leq t\leq T}\Omega_t^{(i)}$. $(e)$ comes from the fact that $\eta\mu < 1$ and $0 \leq \alpha_t^{(i)}(\lambda) \leq 1$ and $0 \leq \Gamma_t^{(i)} \leq 1$ for all $i, t$ and $\sum_{i=0}^t\alpha_t^{(i)}(\lambda) = 1$ for all $t$.

Applying the above result to $T$ global aggregations in the server, we have

$$\mathbb{E}[F(\mathbf{w}_T) - F(\mathbf{w}^*)|\mathbf{w}_0]$$

$$\overset{(a)}{=} \mathbb{E}\left[\mathbb{E}[F(\mathbf{w}_T) - F(\mathbf{w}^*)|\mathbf{w}_{T-1}]|\mathbf{w}_0\right]$$

$$\overset{(b)}{\leq} \mathbb{E}\left[(1 - \gamma + \gamma(1 - \eta\mu)^E)[F(\mathbf{w}_{T-1}) - F(\mathbf{w}^*)]|\mathbf{w}_0\right] + \frac{\gamma(E\eta\rho_1 + 2G_{T-1}(\lambda) + 2\Omega_{max}(E_{th}, \delta))}{2}$$

$$\overset{(c)}{\leq} (1 - \gamma + \gamma(1 - \eta\mu)^E)^T[F(\mathbf{w}_0) - F(\mathbf{w}^*)] + \frac{\gamma(E\eta\rho_1 + 2G_{T-1}(\lambda) + 2\Omega_{max}(E_{th}, \delta))}{2}$$

$$+ \sum_{\tau=1}^{T-1} \frac{\gamma(E\eta\rho_1 + 2G_{T-1-\tau}(\lambda) + 2\Omega_{max}(E_{th}, \delta))}{2}(1 - \gamma + \gamma(1 - \eta\mu)^E)^\tau$$

$$\overset{(d)}{\leq} (1 - \gamma + \gamma(1 - \eta\mu)^E)^T[F(\mathbf{w}_0) - F(\mathbf{w}^*)]$$

$$+ \left[1 - \{1 - \gamma + \gamma(1 - \eta\mu)^E\}^T\right]\frac{E\eta\rho_1 + 2G_{max}(\lambda) + 2\Omega_{max}(E_{th}, \delta)}{2(1 - (1 - \eta\mu)^E)}$$

$$\overset{(e)}{\leq} (1 - \gamma + \gamma(1 - \eta\mu)^E)^T[F(\mathbf{w}_0) - F(\mathbf{w}^*)]$$

$$+ \left[1 - \{1 - \gamma + \gamma(1 - \eta\mu)^E\}^T\right]\frac{\rho_1 + 2\mu G_{max}(\lambda) + 2\mu\Omega_{max}(E_{th}, \delta)}{2\eta\mu^2}$$

$$= \nu^T[F(\mathbf{w}_0) - F(\mathbf{w}^*)] + (1 - \nu^T)Z(\lambda, E_{th}, \delta)$$

which completes the proof. Here, $(a)$ comes from the *Law of total expectation*, $(b)$, $(c)$ are due to inequality (8). $(d)$ is obtained from the definition of $G_{max}(\lambda) := \max_{1 \leq t \leq T} \sum_{i=0}^{t-1} \alpha_t^{(i)}(\lambda)e_t^{(i)}$. In addition, $(e)$ is from $\eta\mu \leq 1$.

## M   Additional discussions on the impact of staleness exponent $\lambda$

From Theorem 1, we confirmed that the error term $G_{max}(\lambda)$ caused by stragglers can be reduced by increasing the staleness exponent $\lambda$. But we also note that choosing a very large $\lambda$ makes our Sagflow to consider only the group with the smallest staleness in each global aggregation; Sagflow reduces to the *ignore stragglers* scheme, which can lose significant data at each round and often converges to a suboptimal point in practice. Fig. 14 below shows the result with varying $\lambda$ using FMNIST dataset. Each device can have delay of 0, 1, 2 which is determined independently and uniformly random. The results indicate that an appropriate $\lambda$ has to be chosen to provide more weights to the recent groups with small staleness, while not totally ignoring the results with large staleness.

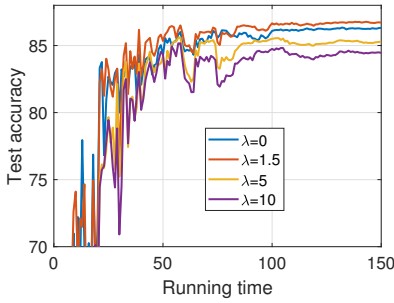

Figure 14: Performance of Sageflow with different $\lambda$ values considering only stragglers. An appropriate $\lambda$ has to be chosen to provide more weights to the recent groups with small staleness, while not totally ignoring the results with large staleness.

**Comparison with FedAsync [7]:** To compare our Sagflow with FedAsync, we consider a setup with only stragglers. Let $n_t = \sum_{i=0}^{t} |U_t^{(i)}|$ be the number of models received at global round $t$. Note that our scheme performs staleness-aware grouping with those $n_t$ models to update the global model

once, while FedAsync updates the global model $n_t$ times one-by-one. Hence, after $T$ global rounds, FedAsync performs $c(T) = \sum_{t=1}^{T} n_t$ updates at the server. If we apply the same proof technique of ours, we have $\mathbb{E}[F(\mathbf{w}_{c(T)}) - F(\mathbf{w}^*)] \leq \nu^{c(T)}[F(\mathbf{w}_0) - F(\mathbf{w}^*)] + (1 - \nu^{c(T)})Z$ for FedAsync where $\nu = 1 - \gamma + \gamma(1 - \eta\mu)^E$,

$$Z = \frac{\rho_1 + 2\mu G_{max}}{2\eta\mu^2}, \tag{9}$$

$$G_{max} = \max_{0 \leq c \leq c(T), 0 \leq i \leq c-1} [F(\mathbf{w}_i) - F(\mathbf{w}_c)]$$
$$= \max_{0 \leq c \leq c(T), 0 \leq i \leq c-1} e_c^{(i)}. \tag{10}$$

Note that FedAsync controls $\gamma$ based on the staleness, which controls $\nu$. However, the staleness exponent of FedAsync does not control the $G_{max}$ term of (10) directly. Compared to FedAsync, our $G_{max}$ term (9) can be directly controlled by staleness exponent $\lambda$, which affects $\alpha_t^{(i)}(\lambda)$; by choosing an appropriate $\lambda$, we can reduce the errors caused by stragglers with larger staleness in $G_{max}$. Regarding the global updates, it can be seen that more global updates are performed in FedAsync than Sageflow under the same conditions. However, in order to reduce the error term $G_{max}$ in FedAsync, $\gamma$ should be reduced which makes convergence speed slower. Compared to FedAsync, our staleness-aware grouping can keep $\gamma$ high while reducing $G_{max}$, by controlling the staleness exponent $\lambda$.

## N  Additional discussions on the impact of $E_{th}$ and $\delta$

As we stated in the main manuscript, we can filter out the adversaries with high entropies by choosing an appropriate $E_{th}$. Here, we note that selecting a small $E_{th}$ can degrade the performance of Sageflow (as in Fig. 4) since not only the adversaries but also the benign devices are filtered out with a large $E_{th}$. As can be seen from Fig. 4, $E_{th}$ is a hyperparameter that can be easily tuned since there is a huge gap between the entropy values of benign versus adversarial devices.

Regarding $\delta$, it can be easily seen that the error term caused by adversaries goes to 0 as $\delta$ increases: for an adversary device $k \in M_t^{(i)}(E_{th})$, we can write

$$\beta_i^{(k)}(\delta)[F(\mathbf{w}_t(k)) - F(\mathbf{w}^*)] \leq \frac{\frac{m_k}{F_{pub}(\mathbf{w}_t(k))^\delta}}{\sum_{j \in U_t^{(i)}(E_{th})} \frac{m_j}{F_{pub}(\mathbf{w}_t(j))^\delta}} F(\mathbf{w}_t(k))$$
$$\leq \frac{\frac{m_k}{F_{pub}(\mathbf{w}_t(k))^\delta}}{\frac{m_{j_B}}{F_{pub}(\mathbf{w}_t(j_B))^\delta}} F(\mathbf{w}_t(k))$$
$$= \frac{m_k}{m_{j_B}} \frac{F_{pub}(\mathbf{w}_t(j_B))^\delta}{F_{pub}(\mathbf{w}_t(k))^\delta} F(\mathbf{w}_t(k)) \tag{11}$$

where $j_B$ is an arbitrary benign device chosen from $U_t^{(i)}(E_{th})$. Here, since $\mathbf{w}_t(j_B)$ is the model of a benign device, we can write $F_{pub}(\mathbf{w}_t(k)) \gg F_{pub}(\mathbf{w}_t(j_B))$ for an adversarial device $k$ under data poisoning or scaled backdoor attacks. Therefore, we can conclude that $\Omega_t^{(i)}(E_{th}, \delta) = \sum_{k \in M_t^{(i)}(E_{th})} \beta_i^{(k)}(\delta) F(\mathbf{w}_t(k)) - F(\mathbf{w}^*)]$ goes to 0 as $\delta$ increases. However, the effect of adversaries can be sufficiently reduced even with a fixed $\delta = 1$, as can be seen in the experiments in the main manuscript and Supplementary Material. We also note that selecting a very large $\delta$ degrades the performance of Sageflow as observed in Fig. 4. This is because only the device having the smallest loss is considered with a very large $\delta$, ignoring the effects of other benign devices. As shown in the results in Fig 4, $\delta$ is a hyperparameter that can be easily tuned since the models of adversaries have relatively large losses under data poisoning or scaled backdoor attacks.

To sum up, we have to choose an appropriate $E_{th}$ and $\delta$ to achieve a desired level of performance, which is easy; the performance of Sageflow is not highly sensitive to those hyperparameters as long as they are chosen in a reasonable range as shown in Fig 4.