# OpenReview forum: "Sageflow: Robust Federated Learning against Both Stragglers and Adversaries"
_NeurIPS.cc/2021/Conference — NeurIPS 2021 Poster_

### Official Review · Reviewer_voGc · 2021-07-17

**Rating:** 6
**Confidence:** 3

**Summary:**

The paper addresses a scheme to protect federated learning against stragglers and adversaries at the same time. For straggler resistance, the authors propose to group and weight the model coefficients according to their age. For resistance against adversaries they propose entropy-based filtering and loss-weighted averaging. Combining these schemes leads to their proposed Sageflow approach. They also provide an upper bound on the expected loss function error after a fixed number of operations.

**Limitations And Societal Impact:**

Yes.

**Main Review:**

Although the paper is merely a combination of two different schemes, which seems to be orthogonal in its impact, the strengths of the paper are in my opinion the convincing numerical results compared to other schemes and the convergence analysis, which ties the performance limiting impacts of straggling and adversarial nodes together.

On the other hand, the paper has several weaknesses, mostly in terms of clarity. While the description with respect to the straggler resistance was near to me, I was not able to follow their strategy to combat adversarial actions:
- I do not understand the underlying model to eq.(4) and why publicly available data helps. Specifically, how is P^{(q)}_{x_pub} computed? I must be fundamentally misunderstanding the idea as we do have these training examples the corresponding labels available. So isn't the corresponding probability always 1? The authors need to clarify this point in their response.
- In line 180, why is the loss function computed between the weights and the data x? This does not make any sense to me.
- The authors should also clarify the novelty of their approach. It is clear that perhaps the combination and the bound in Theorem 1 is new, but what about the individual approaches with respect to stragglers and adversaries?
- Gradient coding is another method to provide straggler resistance in synchronous federated learning. It would be useful to compare their straggler approach with these synchronous schemes to properly assess the merit of their work.

If the authors satisfactorily respond to these questions (in particular to my first point), I am willing to increase my rating.

---------------------

After the authors's response:
From reading the authors'c comments to my questions, the other reviews, and their response to those, I understand their scheme better now. Consequently, I am updating my score.

**Time Spent Reviewing:**

8

---

> ### Author Response · Authors · 2021-08-08
> **Response to Reviewer voGc**
>
> We appreciate the reviewer's comments on our paper. Our answers to the reviewer's questions are given as below:
>
> 1. **Computing entropy in equation (4):** $P_{x_{pub}}^{(q)}$ is the softmax output corresponding to the $q$-th class ($q$-th output of the softmax classifier) when the input of the model is $x_{pub}$. Hence, $P_{x_{pub}}^{(q)}$ is generally a non-zero value for each $q$ and we have $\sum_{q=1}^QP_{x_{pub}}^{(q)}=1$. Now based on $P_{x_{pub}}^{(q)}$, the entropy can be computed as in line 158 using the log function, and this entropy value is maximized when $P_{x_{pub}}^{(q)}=1/Q$ for all $q$. When the model is trained well on a specific dataset, the model produces a high-confident prediction for the ground truth labels of the trained classes and thus has a low entropy for the prediction. However, the models compromised by adversaries (model poisoning) tend to predict randomly for all classes and thus have high entropies compared to the models of benign devices. Fig. 2 in our main manuscript confirms this intuition. This is the reason why the entropy with the public data in equation (4) works.
>
> 2. **Computing the loss in line 180:** The loss in line 180 is exactly the conventional loss (e.g., cross-entropy loss) which is computed when the model and the data sample is given: given a sample x, we compute the output of the model w after forward propagation, and then compute the cross-entropy loss using the corresponding label. We will make this clearer by introducing the label y in the equation; thanks for the comment.
>
> 3. **Novelty of individual approaches:** As we stated in the current manuscript, our “staleness-aware grouping” (targeting stragglers) and “entropy-based filtering and loss-weighted averaging” (targeting adversaries) are new, and have their own novelty in terms of both algorithm and performance: first, to the best of the authors’ knowledge, staleness-aware grouping is the first work that adopts group-aggregation depending on the staleness factor. As can be seen in Fig. 3 with only stragglers, the performance of staleness-aware grouping is not only effective in mitigating the effect of stragglers but also provides a great platform for countering adversaries, as shown in Fig. 5 with both stragglers and adversaries. Our entropy-based filtering and loss-weighted averaging is also a new approach to handle adversaries. As can be seen in Fig. 4 with only adversaries, entropy-based filtering and loss-weighted averaging work in a highly complementary fashion to effectively counter a wide range of adversarial attacks. To sum up, the results in Fig. 3 and Fig. 4 confirm the advantage of the two individual methods, while Fig. 5 with both stragglers and adversaries confirms the synergy effect of combining these two unique/novel ideas. As illustrated in Fig. 5, if we adopt only one of these methods and combine with an existing idea, the performance is significantly degraded, which confirms the synergy effect of combining our new ideas. The reason why these two ideas show a better synergy effect than other combinations is illustrated throughout the introduction and experimental results.
>
> 4. **Gradient coding:** We stress that gradient coding can be only utilized in a distributed learning setup (not in a federated learning setup) where the overall dataset is collected at the central server and multiple nodes are utilized for data parallelization: the idea of coding is applied when the central server distributes the collected data to multiple nodes. However, since each node collects its own dataset in federated learning, which is our focus, various coding-based ideas including gradient coding cannot be directly applied. Please note that we already considered various common baselines to handle stragglers, including FedAsync, ignore stragglers, wait for all stragglers, and wait for partial stragglers.
>
> Thanks again for your time and efforts in reviewing our paper. Please let us know if there are any additional questions that we should address.
>
> Best, Authors

---

> ### Author Response · Authors · 2021-08-18
> **Dear Reviewer voGc**
>
> Dear Reviewer voGc
>
> Once again we appreciate your willingness to reevaluate your score given satisfactory explanations on the points you raised. We feel we did our best in trying to make our answer to your questions as clear as possible. If there are any more clarifications we could provide, we would be grateful if you could let us have an opportunity.
>
> Best Wishes.
>
> Authors

---

### Official Review · Reviewer_BUwC · 2021-07-18

**Rating:** 7
**Confidence:** 4

**Summary:**

This paper proposes an approach, Sageflow, that jointly handles stragglers and Byzantine users in FL. Sageflow groups the users based on their staleness to handle stragglers and uses entropy-based filtering and loss-weighted averaging to provide Byzantine-robustness. This approach depends on the assumption that the server has access to a public dataset or the users share part of their data with the server to to identify the Byzantine users. Several experiments show that effectiveness of the Sageflow against some of the prior approaches.

**Limitations And Societal Impact:**

There is no discussion on the limitations and potential negative societal impact. Since the work focuses on Byzantine mitigation, it has a positive societal impact. Please also see my comments above for my suggestions.

**Main Review:**

The proposed approach, Sageflow, handles stragglers and Byzantines jointly and prior works only focus on one of these two issues.
The idea of adapting the weights while aggregating the models based on the staleness is a nice idea as the the updates get less important as they become stale. Sageflow also combines between entropy-based filtering and loss-based averaging to handle Byzantine users, which is a good idea as each metric is more suitable for certain class of attacks.

I overall like the proposed approach, but there are some issues that should be addressed as I discuss below.

1- There are some prior approaches that also use data sharing to handle Byzantine users such as [1, 2], but this paper does not compare with them.

While these works only deal with Byzantine users and do not consider stragglers, it is good to compare the proposed strategy with them.
You should also consider comparing with the OracleSGD [2] as a reference.

2- Line 124: "Note that the result sent from device $k \in U_t^{(i)}$ is the model after E local updates starting from $\mathbf w_i$".

3- The entropy-based filtering requires performing inference using the models of the users and the public samples at the servers. This seems significant and there's no discussion on the complexity of doing so. What's the complexity of this?

4- I do not see why high entropy always means that the user in Byzantine, what if this is due to the non-IID data distribution?
It's mentioned later in line 162 that some attacks tend to make the entropy high as they make the predictions random, but some other attacks also may be biased toward specific class and hence they result in a low entropy.

5- Line 139: It's mentioned that the active users who get a new request should ignore this request. It may be better for those users instead to respond to this request and ignore the computations that they are doing. This is intuitive as the fresh updates are more important than the stale updates. Why ignoring the new request is better?

6- Line 170: You mention that your work can handle large number of adversaries, what's this number? and how do you decide the threshold for the entropy-based filtering?

In Multi-Krum, 2f+2<N, what about Sageflow?

7- A limitation of this work also is that it does not consider the convergence for the non-convex case.
This seems to be doable as a recent work considering a similar problem has done this analysis.

This work appeared after the deadline for NeurIPS, so it's not an issue at all that you have not considered it.

8- Why did you ignore the batch normalization layers in Section 4? Please explain and add results with batch normalization as well.

9-  Line 86: You mention independent, identically distributed, but this should have been mentioned the first time the term IID is used.

10- Line 292 has a typo: "percantage"

[1] X. Cao et al. "FLTrust: Byzantine-robust Federated Learning via Trust Bootstrapping." arXiv preprint arXiv:2012.13995 (2020).

[2] S. Prakash, and A. S. Avestimehr. "Mitigating byzantine attacks in federated learning." arXiv preprint arXiv:2010.07541 (2020).

[3], J. Nguyen et al. "Federated Learning with Buffered Asynchronous Aggregation." arXiv preprint arXiv:2106.06639 (2021).



**Time Spent Reviewing:**

3-4 hours

---

> ### Author Response · Authors · 2021-08-09
> **Response to Reviewer BUwC**
>
> We appreciate the reviewer for the feedback and insightful comments. We will give responses to the suggestions by the reviewer.
>
> 1. **Comparison with more previous works:** We appreciate the reviewer’s comment. We performed additional experiments by comparing our scheme with FLTrust [1] and OracleSGD [2]. The setup is exactly the same in Fig. 5 of our main manuscript. We considered model poisoning attack with scale 0.1. To this end, we applied FLtrust and OracleSGD in each grouping stage of our staleness-aware aggregation. Since FLTrust also utilizes public data, we allocated the same public data for FLTrust as Sageflow. For OracleSGD, we performed FedAvg using the models of the benign devices. Note that the performance of OracleSGD can be viewed as an ideal performance as it assumes the full knowledge on which devices are the adversaries. When public data is class-balanced (the number of samples are distributed uniformly across the classes in the public data), our scheme achieves the accuracy of 86.54% at running time 120 on FMNIST while 86.56%, 86.91% are achievable for FLTrust, OracleSGD, respectively. However, under the setting of Fig. 11 in Supplementary Material (when the public data is class-imbalanced), our scheme achieves accuracy of 85.6% at running time 120 on FMNIST while 81.5%, 86.91% are achievable for FLTrust, OracleSGD, respectively. Now using CIFAR10 with a class-balanced public data, our scheme and OracleSGD achieve 66.48% and 66.05% respectively, while FLTrust does not work well (achieving accuracy of 10%) on this relatively complicated dataset under our severe non-IIDness scenario. The overall results confirm the advantage of Sageflow in various practical settings. We will take these baselines into account in the revised manuscript as the reviewer suggests.
>
> [1] X. Cao et al. "FLTrust: Byzantine-robust Federated Learning via Trust Bootstrapping." arXiv preprint arXiv:2012.13995 (2020).
>
> [2] S. Prakash, and A. S. Avestimehr. "Mitigating byzantine attacks in federated learning." arXiv preprint arXiv:2010.07541 (2020).
>
> 2. **Complexity of Sageflow:** Although the complexity of our scheme is discussed in Lines 203-206 of the current manuscript, we try to make it clearer as follows: let K be the number of devices participating in each global round and $|w|$ be the number of model parameters. Also let $n_{pub}$ be the number of public data samples. Assuming that the time complexity of computing the entropy or loss is linear in $|w|$ (as in [Xie19]), the time complexity of our scheme can be written as $O(n_{pub}|w|K)$, which scales linearly with $K$. The time complexity of Zeno (which also utilizes public data to handle adversaries) is also $O(n_{pub}|w|K)$, the same as ours, but suffers from significant performance degradation as can be seen in Figs. 4 and 5 of the main manuscript. In contrast, the time complexity of Krum (which is a well-known method to handle adversaries) is $O(|w|K^2)$, which does not scale linearly with the number of participants $K$.
>
> [Xie19] C. Xie et al., "Zeno: Distributed stochastic gradient descent with suspicion-based fault-tolerance." ICML 2019.
>
> 3. **Providing more insights on entropy:** We thank the reviewer for the comment. As can be seen in Fig. 2(a) of the main manuscript, under specific model poisoning attacks (like reverse sign attack), the models compromised by adversaries tend to predict randomly for all classes and thus have high entropies compared to the models of benign devices. Even though a client has a biased dataset (due to the non-IID data distribution), the entropy of the benign devices tends to be significantly lower than the entropy of the adversaries, under model poisoning attacks (see Fig. 2(a) in the main manuscript and Fig. 9 in the Supplementary Material for the details). Hence, we can easily filter out only the adversaries, not the benign devices, via entropy. However, as per the reviewer’s comment, there are cases where entropy cannot play a role since the attack makes the model to be biased toward specific classes. Data poisoning is one example of such attack. In such cases, the models of adversaries generally produce high loss values and thus loss-weighted averaging can play a key role as in Fig. 2(d) of the main manuscript. This is the reason why we consider both entropy-based filtering and loss-weighted averaging simultaneously, which work in a highly complementary fashion to counter a wide range of adversary attacks.
>
> 4. **Why ignore new request?** When a new request comes, there are two possible actions that an active device can take: ignoring the new request (as in the current manuscript) or stop the current computation and start the new one (as per the reviewer’s comment). We took the former idea since each client can eventually contribute to the global model within few global rounds with this method. For the latter method, if a specific straggler is selected to be the active node multiple times, the current computational result is continuously discarded and the data samples of that device cannot contribute to the global model for a long period of time.
>
> 5.  **Number of adversaries that Sageflow can handle:** Similar to the geometric-median-based schemes such as RFA (and different from Multi-Krum), the number of adversaries that our scheme can combat with is not characterized in a closed-form: the performance is gradually degraded as the portion of adversaries increases as RFA. However, there is a big difference between our scheme and RFA: while RFA does not work at all when the portion of adversaries is above 50%, our scheme can still achieve a reasonable performance as can be seen in Fig. 12 of the Supplementary Material. This is because geometric-median-based methods are significantly more sensitive to the portion of adversaries (for taking the geometric median) compared to our entropy-based filtering and loss-weighted averaging. The threshold for the entropy-based filtering can be easily tuned since there is a huge gap between the entropy values of benign versus adversarial devices as in Fig. 2. Please refer to Section D of the Supplementary Material for more details on selecting the entropy threshold value.
>
> 6. **Convergence for the non-convex case:** We appreciate the reviewer’s comment.
>
> 7. **Batch normalization:** We ignored the batch normalization as in the previous works [Hsieh20], [Wang20] because batch normalization often does not work well in federated learning setups with non-IID data distributions, as described in [Hsieh20]. It is shown in [Hsieh20] that other methods such as group-normalization can handle this issue. More recently, the authors in [Li21] analyzed the effect of batch normalization in federated learning and proposed another method to handle this issue. Having said that, we performed additional experiments with batch normalization and obtained consistent results: under the same setting in Fig. 5 with both stragglers and adversaries, our Sageflow achieves 64.4% accuracy while 54.36%, 26.81%, 10% and 42.64% are achieved for FedAsync + eflow, Sag + RFA, Ignore stragglers + RFA, Wait for stragglers + RFA respectively.
>
> [Hsieh20] K. Hsieh et al. "The non-iid data quagmire of decentralized machine learning." ICML 2020.
>
> [Wang20] H. Wang, et al. "Federated learning with matched averaging," ICLR 2020.
>
> [Li21] X. Li et al. "Fedbn: Federated learning on non-iid features via local batch normalization," ICLR 2021.
>
> 8. **Defining the term IID:** We will define the term IID in an appropriate location.
>
> 9. **Typo:** We will correct the typo accordingly.
>
> 10. **Limitation and societal impact:** Thanks for the comment; we will add something to the effect: Our Sageflow requires a minimum amount of public data at the server; when the public data related to the task is not available at all (e.g., in financial applications), it is hard to utilize Sageflow. Regarding societal impact, stragglers with large staleness would provide only minor contributions to the global model, which can lead to a fairness issue.
>
> We again appreciate the reviewer for the time and efforts. Please do let us know if you have any further comments/questions.
>
> Best, Authors

---

> > ### Comment · Reviewer_BUwC · 2021-08-12
> > **DiverseFL & Non-Convex Convergence**
> >
> > I would like thank you for the additional experiments that you have performed. However, there are two of my comments that are not addressed completely.
> >
> > 1) How does your work compare to DiverseFL[2]?
> > This needs more experiments. You only compared with the OracleSGD scheme suggested in this work, but I do not see any comparison with DiverseFL.
> >
> > 2) Why it is not possible to extend your analysis to the non-convex case? I said it's not a problem that you did not consider and did not mention [3], but that does not mean you should not  consider the non-convex case.

---

> > > ### Author Response · Authors · 2021-08-12
> > > **We appreciate the comments**
> > >
> > > We really appreciate the reviewer for further comments on our paper. We will try our best to answer to the specific questions as soon as possible.
> > >
> > > Best, Authors

---

> > > ### Author Response · Authors · 2021-08-13
> > > **Response to Additional Comments**
> > >
> > > We appreciate the reviewer for providing additional constructive comments on our paper. Our answers are given as below:
> > >
> > >  **Comparison with DiverseFL:** We now performed additional experiments using DiverseFL, under the same setup as in Fig. 5 with both stragglers and adversaries. We utilized DiverseFL in each grouping stage of our staleness-aware grouping to compare with our Sageflow. For a fair comparison, we let each client to send 2\% of its local dataset to the server in DiverseFL. The server utilizes each of the received “local dataset” to compute the gradient of each client. Then, the server computes the similarity between each of the computed gradient and the received gradient sent from the corresponding client, to filter out adversaries. For MNIST, our scheme achieves accuracy of 97.71\%, while DiverseFL achieves 97.58\%. For FMNIST, the accuracies are 86.54\% and 85.55\% for our scheme and DiverseFL, respectively. Finally, 87.89\% and 85.94\% are achieved for our scheme and DiverseFL using FEMNIST dataset.
> > >
> > > Here we note that DiverseFL requires several additional constraints to be utilized in practice compared to our Sageflow. First of all, in order for the server to compute the similarity between gradients, it is essential for the clients participating in FL to directly send their local data to the server, which may cause privacy issue. A more recent version of this paper (uploaded on arXiv in July 2021) creates a Trusted Execution Environment (TEE) inside the server targeting this privacy issue. Secondly, the adversaries may upload the corrupted data to the server, which makes DiverseFL challenging to combat adversaries. Although the authors of this paper claim that the group of experts can remove this corrupted data at the server, this requires additional resources and efforts. Moreover, in order for the server to compute the gradient of the clients, DiverseFL requires the server to distinguish the local datasets of all clients. In other words, the server should remember which dataset come from which client, which  can be challenging in cross-device FL scenarios having a significant number of clients in the system. Finally, at the server-side, DiverseFL needs to perform a large number of forward/backward propagations for computing the gradient (after multiple updates) of all clients in each global round, which causes additional computational burden at the server.
> > >
> > > **Convergence analysis for the non-convex case:** We first apologize for misunderstanding the reviewer’s previous comment. We would like to say that the problem setup and the algorithm of our work make the analysis for the non-convex case more challenging: we consider both stragglers and adversaries simultaneously, introduce the staleness exponent $\lambda$ to aggregate the group representative models, filters out and reduces the impact of adversaries by introducing entropy threshold $E_{th}$ and loss exponent $\delta$, which have not been considered before. We would like to stress that this is totally different from the scheme of [3] even when only stragglers exist, since the scheme of [3] directly aggregates the gradients of the clients without considering the staleness factor. In our scheme, staleness factor is taken into account when aggregating the group representative models. As illustrated in [3] that the reviewer suggested (page 3, line 5), considering the staleness factor makes the analysis complicated and challenging.
> > >
> > > Nevertheless, we took a look at [3], and tried to figure out how this result can be related to our work. Consider a setup with only stragglers as in [3] with $E_{th}=\infty$ and $\delta=0$. Under this setup, we see that **our scheme with $\mathbf{\lambda=0}$** is very similar to the scheme in [3], since the staleness effect is not considered with $\lambda=0$ in our scheme. The difference is that our scheme aggregates the model periodically, while the scheme in [3] aggregates the model when a fixed number of models are received at the server. Assume that the server aggregates the models after receiving $K$ results as in [3]. We also let the time-average coefficient of our scheme as $\gamma=0$. In this special case with $\lambda=0$, $E_{th}=\infty$, $\delta=0$, the convergence rate can be obtained following the same procedure of the proof of Theorem 1 of [3].  Using our notation, the following convergence rate is achieved:
> > > $$
> > > \frac{1}{T}\sum_{t=0}^{T-1} \mathbb{E}[||\nabla F(\mathbf{w}_T)||^2] \leq \frac{2(F(\mathbf{w}_0) -F(\mathbf{w}^*) )}{E\eta T} + 3L^2E^2\eta^2(\tau^2 +1)(\rho_1 + \rho_2 ) + \frac{L\eta \rho_1}{2K}
> > > $$
> > > which shows the same behavior as the result in [3]. Here, $\rho_2$ is the upper bound of the gradient norm of the local loss function, i.e., $\|\nabla F_k\|^2\leq \rho_2$ for all $k$. $\tau$ is the maximum value of the  staleness (the difference between the current round and the past round when the device received the model) that a client can have at each global round. The first term of the right-hand side goes to zero as global round $T$ increases, while the other terms can be viewed as the error terms. We will clearly describe the challenge and relation to [3] in the revised manuscript.
> > >
> > > Best, Authors

---

> > > > ### Comment · Reviewer_BUwC · 2021-08-14
> > > > **Convergence Analysis**
> > > >
> > > > Thanks for your efforts. The answer to the convergence analysis is unsatisfactory, but the problem is challenging. However, I updated my score since you added the experiments I asked for.

---

> > > > > ### Author Response · Authors · 2021-08-14
> > > > > **We appreciate the reviewer for the efforts**
> > > > >
> > > > > Thank you for your time and efforts, and acknowledging our contributions.
> > > > >
> > > > > Best, Authors

---

> > ### Comment · Reviewer_BUwC · 2021-08-31
> > **Privacy Limitations**
> >
> > I think the staleness-aware grouping strategy may lead to a privacy leakage. This is because you may have a group with only one user for instance. In this case, secure aggregation cannot help in ensuring privacy.  In general, the privacy guarantee will be limited by the size of the smallest group. If the smallest group has small number of users, the server may estimate the local model of the users in this group. It is worth mentioning this limitation in the paper.

---

> > > ### Author Response · Authors · 2021-08-31
> > > **Response to the privacy issue**
> > >
> > > Yes, as you noted, there can be an issue when there is only one device in a group (secure aggregation guarantees perfect secrecy for two or more devices in a group). The single-device case can be handled based on the **private buffer** in [arXiv'21], the paper that you suggested in the initial review: If a specific group has only one device, one can simply discard that group, or aggregate that specific device with the devices in the adjacent group having the most similar staleness, inside the private buffer. We will clearly mention this in our revised manuscript. Thank you for your feedback!
> > >
> > > [arXiv'21] Federated Learning with Buffered Asynchronous Aggregation, arXiv preprint arXiv:2106.06639, 2021.

---

> > > > ### Comment · Reviewer_BUwC · 2021-08-31
> > > > **Privacy Limitations**
> > > >
> > > > 1- The guarantee of secure aggregation depends on the number of models being aggregated. When only two models are aggregated for instance, the server can estimate these local models through their average. This can be a good estimate especially in the IID setting.
> > > >
> > > > 2- The private buffer solution you suggested based on [arXiv'21] is not desirable in any case as it depends on trusted execution environments (TEEs), which are slow and have a limited memory. It is also not clear how this will affect the performance of your algorithm, since you deal with a given model as if it is received later or earlier.
> > > >
> > > > In any case, it is important to mention this limitation in the paper explicitly.

---

> > > > > ### Author Response · Authors · 2021-09-01
> > > > > **Thanks for the feedback**
> > > > >
> > > > > Thank you for your time and efforts. We will clearly mention the related issue and possible remedies.

---

### Official Review · Reviewer_QeNU · 2021-07-22

**Rating:** 6
**Confidence:** 3

**Summary:**

This paper proposes a robust FL approach to deal with stragglers and adversaries in federated learning. Model grouping and staleness dependent weighting of models is done at the server to deal with stragglers. For model poising attacks, entropy-based filtering is proposed, where a client model entropy is calculated on a public dataset stored at the server. Same dataset is used to calculate loss of different client models at the server to identify clients contributing data-poisoned models. Theoretical bounds on convergence of the proposed method are obtained. Experimental results with three image classification datasets show that the proposed approach outperforms existing solutions to deal with stragglers and adversaries in FL.

**Limitations And Societal Impact:**

The authors have not discussed limitations of their work. You need to provide example of scenarios where your approach may not work. Since your approach requires availability of public data at the server which is representation of global data, your approach will not work in FL settings where such data is not available or no knowledge about global data distribution is available .

**Main Review:**

The paper has some novel ideas, in particular the idea of using both entropy-based filtering and loss-weighted averaging provides protection against both model poisoning and data poisoning attacks. The proposed approach Sagerflow integrates techniques for straggler mitigation and adversarial robustness in FL. The work is technically sound and performance of Sageflow is much better against the baselines.
The paper is well written and the main ideas are explained clearly.
Major comments:
1. Line 39-40: "The presence ...aggregation schemes". This sentence is not clear. What if attackers are also the stragglers ? In that case avoiding stragglers will drive the attack ratio lower. Right ?
2. The assumption that you have access to public data at the server is very strong. First you do not know what is the distribution of the data across all the clients. What should be the data distribution of this public data ? You are using this public data for entropy and loss calculations of individual client models to identify if they are potential adversaries or not. Basically if the entropy is high or the loss is high you mark the device as adversarial. But entropy and loss can be higher every for benign devices if their local data distribution is very different than data distribution at other clients.
3. Higher entropy may not always mean an adversary. Since you are calculating entropy using public data, it may happen that higher entropy of a client model with public data is simply because the client has a very different data distribution than the public data. If this is the case then  filtering out the client model and not using it in global model update is not a good idea. Your global model may never be able to learn from the data local to the client.
4. You are accounting for all the stragglers in your update, even the one which arrive very late. I am not sure if this is a good idea as too stale weights will only make the training unstable. Why not just set a threshold on stragglers even for your grouping approach ?
5. How does Sageflow computational overhead at the server scales with number of clients ? With more clients you need to calculate entropy and loss for more number of clients using the public data. You should show scalability of Sageflow with increasing number of clients .
6. Why were batch normalization layers ignored when training VG-11 and CIFAR10 ?
7. Line 283: What is parameter C ?
8. Your results in the main paper are for very low staleness distribution (0,1,2). How does Sageflow perform with higher sateness values ? What is the final accuracy with Sageflow vs baselines in Figure 3. I would assume the benefits with Sageflow will diminish with high staleness and increased non-iid-ness of data distribution across clients.
9. Use a better term instead of "model update poisoning".

**Time Spent Reviewing:**

4

---

> ### Author Response · Authors · 2021-08-09
> **Response to Reviewer QeNU**
>
> We thank the reviewer for their efforts as well as their valuable comments. Below, we address the comments that the reviewer raised.
>
> 1. **Attack ratio:** As the reviewer commented, the portion of adversaries can become either higher or lower in each global round, depending on the fraction of adversaries that become stragglers. Once the attack ratio becomes higher at a specific global round, the performance of the model is significantly degraded in the current global round, which also leads to a degraded performance in future global rounds. To avoid confusion, we will change our original statement to “The presence of stragglers can drive the attack ratio higher (e.g., by ignoring stragglers),…”.
>
> 2. **Data distribution of public data:** We thank the reviewer for the comment. In fact, we already confirmed (in Figs 4, 5 of the main manuscript and Fig. 9 in Supplementary Material) that Sageflow outperforms existing ideas even when the class distribution of the public data is different from the class distribution across all the clients; even in such cases, the benign devices tend to have smaller entropy/loss values compared to the adversaries, which leads to a successful performance of our Sageflow. We considered both the class-balanced public data (in Figs 4, 5 of the main manuscript) and class-imbalanced public data (in Section G.2 of the Supplementary Material). For both scenarios, it is shown that our Sageflow outperforms existing ideas handling stragglers/adversaries. Now the question is, why Sageflow works well even when the class distribution of the public data does not match with the class distribution across the clients? **This is because the benign devices tend to have relatively smaller entropy or loss values compared to the one with the adversaries, even when the distribution of the public data is different from the data distribution at the clients.** Suppose a benign device has a biased dataset (e.g., only classes 1 and 2), while the public data contains all classes from 1 to 10 (either uniformly or not). Then, the model sent by this benign device has low entropy and the loss for classes 1 and 2 in the public dataset. Here, the overall entropy of this model is lower than the entropy of the model-poisoned device (e.g., reverse sign attack), which produces high entropies for all 10 classes. As can be seen from the results in Fig. 9 of the Supplementary Material, this makes a reasonable gap between the entropy of the benign devices and the adversaries, even when the portion of classes 1 and 2 are small in the public data. Similarly, the overall loss of this model is lower than the loss of the model that is attacked by data poisoning or scaled backdoor attacks, which produces high losses for all 10 classes. This is the reason why entropy-based filtering and loss-weighted averaging work well regardless of the distribution of the public data, and Figs. 2, 4, 5 in our main manuscript and Fig. 9 in the Supplementary Materials support this claim.
>
> 3. **Providing more insights on entropy:** As described in our answer to the second question above, even though a client has a biased dataset (a different data distribution from the public data), the entropy of the benign devices tends to be significantly lower than the entropy of the adversaries, under specific model poisoning attacks (see Fig. 2(a) in the main manuscript and Fig. 9 in the Supplementary Material for the details). Hence, as described in Section D of the Supplementary Material, with an appropriately chosen entropy-threshold, Sageflow enables to filter out only the adversaries (that have significantly larger entropies than the benign devices), not the benign devices. Similarly, the loss values of the benign devices tend to be significantly lower than the loss of the adversaries under data poisoning or scaled backdoor attacks. Fig. 2 in our main manuscript and Fig. 9 in the Supplementary Material support these claims.
>
> 4. **Introducing the staleness threshold:** As the reviewer commented, the server can simply ignore the models if the staleness factor is larger than a certain threshold. However, although this introduces an additional hyperparameter on the threshold value, there is no meaningful performance improvements as described in the following: under the same setting of Fig. 11 of the Supplementary Material with only stragglers and delay of 0 to 8 global rounds, we performed additional experiments with the staleness threshold set to 4. The final accuracy of our Sageflow with a threshold of 4 is 85.67% while the performance with no threshold is 85.53%. The gap is small because the group with a very large staleness already tends to provide only a minor contribution to the global model, due to the effect of the staleness-exponent $\lambda$. The advantage of introducing the staleness threshold can be the reduced computational complexity at the server by ignoring the group of devices with large staleness. We will mention this in our revised manuscript; thanks for the comment.
>
> 5. **Computational overhead:** Let K be the number of devices participating in each global round and $|w|$ be the number of model parameters. Let $n_{pub}$ be the number of public data samples. Assuming that the time complexity of computing the entropy or loss is linear in $|w|$ (as in [Xie19]), the time complexity of our scheme can be written as $O(n_{pub}|w|K)$, which scales linearly with the number of devices $K$. RFA and Zeno also have the same scalability with our scheme. However, for Krum, the time complexity is $O(|w|K^2)$, which does not scale linearly with K. We will try to make this point clearer in the revised manuscript.
>
> [Xie19] C. Xie et al., "Zeno: Distributed stochastic gradient descent with suspicion-based fault-tolerance." ICML 2019.
>
> 6. **Batch normalization:** We ignored the batch normalization as in the previous works [Hsieh20], [Wang20] because batch normalization often does not work well in federated learning setups with non-IID data distributions, as described in [Hsieh20]. It is shown in [Hsieh20] that other methods such as group-normalization can handle this issue. More recently, the authors in [Li21] analyzed the effect of batch normalization in federated learning and proposed another method to handle this issue. Having said that, we performed additional experiments with batch normalization and obtained consistent results: under the same setting in Fig. 5 with both stragglers and adversaries, our Sageflow achieves 64.4% accuracy while 54.36%, 26.81%, 10% and 42.64% are achieved for FedAsync + eflow, Sag + RFA, Ignore stragglers + RFA, Wait for stragglers + RFA respectively.
>
> [Hsieh20] K. Hsieh et al. "The non-iid data quagmire of decentralized machine learning." International Conference on Machine Learning. PMLR, 2020.
>
> [Wang20] H. Wang, et al. "Federated learning with matched averaging," ICLR 2020.
>
> [Li21] X. Li et al. "Fedbn: Federated learning on non-iid features via local batch normalization," ICLR 2021.
>
> 7. **Parameter C in line 283:** As described in line 262, C is the fraction of devices that participate in each global round.
>
> 8. **Results with higher staleness values:** As stated in Line 288 of our manuscript, experiments with larger delays (delay of 0 to 8 global rounds) is already shown in Fig. 11 of the Supplementary Material under the existence of both stragglers and adversaries. The results indicate that our Sageflow still performs better than other schemes in a severe straggler scenario. Regarding the non-IIDness, we stress that we already considered a severe non-IID scenario (throughout all experiments in the current manuscript and the Supplementary Material) by allocating only one or two classes to each client, which makes the local dataset significantly biased. The overall results in Fig. 11 of the Supplementary Material show that our Sageflow outperforms existing ideas under the existence of stragglers (with larger delays) in a strongly non-IID data distribution setup. Regarding the final accuracies in Fig. 3, the achievable accuracies are 68.02%, 67.84%, 65.8%, 63.06%, 10% for Sageflow, FedAsync, Wait for stragglers + FedAvg, Wait for partial stragglers + FedAvg, respectively for CIFAR10 dataset, confirming the advantage of Sageflow.
>
> 9. **Better term instead of model update poisoning:** We agree; we will instead adopt the term “model poisoning”.
>
> 10. **Limitations and societal impact:** A good point. We will add something to the effect: Our Sageflow requires a minimum amount of public data at the server; when the public data related to the task is not available at all (e.g., in financial applications), it is hard to utilize Sageflow. Regarding societal impact, stragglers with large staleness would provide only minor contributions to the global model, which can lead to a fairness issue. **However, we would like to point out that the global data distribution is not required at the server, as we described in the answers to questions 2 and 3 of the reviewer.
>
> Again, we appreciate the reviewer for the time and efforts. Please let us know if you have any other comments/questions.
>
> Best, Authors

---

> ### Author Response · Authors · 2021-08-28
> **Dear Reviewer QeNU**
>
> Dear Reviewer QeNU
>
> Once again we appreciate your time and efforts for providing helpful reviews and raising sharp questions. We feel we did our best in trying to make our answer to your questions as clear as possible. If there are any more clarifications we could provide, we would be grateful if you could let us know.
>
> Best regards,
>
> Authors

---

### Official Review · Reviewer_X8AG · 2021-07-26

**Rating:** 7
**Confidence:** 4

**Summary:**

This paper studies and the interesting problem of presence of both stragglers and adversaries in Federated Learning (FL), and proposes a solution that simultaneously addresses both issues. There's no question that the challenges considered in this paper are important to the FL community. The solution is based on grouping the late model update from stragglers, incorporating them into the model update, and performing an entropy-based filtering to find the adversarial updates. The paper has really interesting and comprehensive experiments, and it is backed by theoretical findings.

**Limitations And Societal Impact:**

The authors do not discuss the limitations and negative societal impact of their work. I suggest authors think deeply about their scheme, and discuss these in the full manuscript. While it may not be very easy to find direct societal impact of this work, topics such as bias and discrimination (when we filter out adversaries, would it be possible that we are also discriminating?) might be easier to discuss. Also, I suggest the authors discuss the limitations of their work, as it would make their paper higher quality and would make it more honest about its assumptions and applicability of the solutions discussed in the paper.

**Main Review:**

The paper proposes an original solution to address both stragglers and adversaries in FL. The idea is novel and very interesting. The claims in the paper are backed by theory and comprehensive experiments. The sheer number of experiments and ablation studies are quite useful in evaluating the claims of the paper. The paper is well-written and it is easy to read

I have the following major comments:
1. Complexity: It seems like the proposed solution has high running time complexity. In particular, the equation (4) is calculated for every update. Computing entropy seems to be of high complexity (I might be wrong, but I get this feeling by reading the paper, as it goes through all data samples in public dataset and also depends on number of classes in dataset). Can the authors explain the asymptotic time complexity of their algorithm? How does the asymptotic time complexity of their algorithm compares to the other related algorithms mentioned in the paper?

2. Datasets: The authors have used the following datasets: MNIST, FMNIST, and CIFAR10. These datasets are not always representative of the challenges in FL. The gold standard benchmark for FL is LEAF: https://leaf.cmu.edu.

3. Question: It would be great if authors could provide more explanation/justification why entropy is a good measure? In the current writeup of the paper, it seems a bit ad-hoc why entropy is a good measure to detect some attacks.

Some minor suggestions:
- when referring to supplemental material from the main text, it would be nice to refer to the exact section.
- some figures can be improve. E.g. fig 2c and fig 6 have data points all over the figure which makes the figures hard to read.


**Time Spent Reviewing:**

7 hours

---

> ### Author Response · Authors · 2021-08-09
> **Response to Reviewer X8AG**
>
> We appreciate the reviewer for the review and the thoughtful comments. Below, we reply to the comments raised by the reviewer.
>
> 1. **Complexity comparison:** We appreciate the reviewer’s comment. Although the complexity of our scheme is discussed in Lines 203-206 of the current manuscript, we add further comments here to make the points clearer: Let $K$ be the number of devices participating in each global round and $|w|$ be the number of model parameters. Also let $n_{pub}$ be the number of public data samples. Assuming that the time complexity of computing the entropy or loss is linear in $|w|$ (as in [Xie19]), the time complexity of our scheme can be written as $O(n_{pub}|w|K)$, which scales linearly with the number of devices $K$. The time complexity of Zeno (which also utilizes public data to handle adversaries) is also $O(n_{pub}|w|K)$, the same as ours, but suffers from significant performance degradation as can be seen in Figs. 4 and 5 of the main manuscript. On the other hand, the time complexity of Krum (which is a well-known method for handling adversaries) is $O(|w|K^2)$, which does not scale linearly with $K$. We will make this point clearer in the revised manuscript.
>
> [Xie19] C. Xie et al., "Zeno: Distributed stochastic gradient descent with suspicion-based fault-tolerance." ICML 2019.
>
> 2. **Datasets for FL:** Based on the reviewer’s comment, we performed additional experiments using FEMNIST (one of the datasets in LEAF) and obtained consistent results: under the same setting of Fig. 5 in the main manuscript with both stragglers and adversaries (model-update poisoning), our scheme achieves accuracy of 86.78% at running time 100 while 80.19%, 76.17%, 10%, 10% are achievable for FedAsync + eflow, Sag + RFA, Ignore stragglers + RFA, Wait for stragglers + RFA, respectively. We believe that these results further confirm the advantage of our scheme compared to various baselines.
>
> 3. **Providing more insights on entropy:** As described in Lines 160-165 of the main manuscript, our intuition is that in some attacks, the models compromised by the adversaries would tend to predict randomly for all classes and thus have high entropies compared to the models of benign devices. Even when the local dataset of the benign device is biased, it would have high confidence (i.e., low entropy) on the classes that are in their local dataset, making the entropy lower compared to the model of the adversaries. Model poisoning attack is one example that shows this behavior, which was confirmed via experiments in Fig. 2 of our current manuscript. We note that the inflicted models cannot be filtered out based on the loss values in the setting of model poisoning in Fig. 2 (see Fig. 2(b)), which confirms the importance of the role of entropy. However, there are some cases where entropy cannot play a role, e.g., when the attacker makes the model to be biased toward specific classes. Data poisoning is one example of such attack. In such cases, the models of the adversaries generally produce high loss values and thus loss-weighted averaging can play a key role. This is the reason why we consider both entropy-based filtering and loss-weighted averaging simultaneously, which work in a highly complementary fashion to counter a wide range of adversary attacks as shown in Fig. 2 of the main manuscript. We will try to make these points clearer in the revised manuscript.
>
> 4. **Referring to Supplementary Material and improving figures:** We will revise accordingly.
>
> 5. **Limitations and societal impact:** Thanks for the suggestion. We will add something to this effect: Our Sageflow requires public data (albeit a small amount) at the server; when the public data related to the task is not available at all (e.g., in financial applications), it is hard to utilize Sageflow. Regarding societal impact, stragglers with large staleness would provide only minor contributions to the global model, which can lead to a fairness issue.
>
> Again, thank you for your valuable comments. Please do let us know if you have any remaining questions/comments.
>
> Best, Authors

---

> ### Comment · Reviewer_X8AG · 2021-08-25
> **Thanks for the response**
>
> I thank the authors for providing responses to my questions.
>
> All of my questions are answered and I am satisfied with the responses. I also read the responses to other reviewers, and found them very useful in answering my doubts. I found all of them really useful and detailed. Thanks!
>
> The last question/concern I have (but I am also not sure about it), is the following claim in the response to a reviewer: "However, the models compromised by adversaries (model poisoning) tend to predict randomly for all classes and thus have high entropies compared to the models of benign device".
> Is this true? Are we saying that model poisoning does random guessing for classes? Can we imagine a scenario where the adversary produces a non-random high-confident prediction and thus has a low entropy for the prediction?

---

> > ### Author Response · Authors · 2021-08-25
> > **Additional Response to Reviewer X8AG**
> >
> > Yes, this is what we have observed for model poisoning (reverse sign attack), as can be seen in Fig. 2(a) of the main manuscript; the models compromised by adversaries (model poisoning) tend to have the maximum entropy value. However, as per the reviewer's comment (and also Reviewer BUwC commented), there are some other attacks that make the model to be biased toward specific class and thus resulting in a low entropy. Data poisoning is one example of such attack, as can be seen in Fig. 2(c). In such cases, the models of the adversaries generally produce high loss values and thus **loss-weighted averaging can play a key role** (see Fig. 2(d)). This is the reason why we consider both entropy-based filtering and loss-weighted averaging simultaneously, which work in a highly complementary fashion to counter a wide range of adversary attacks as shown in Fig. 2 of the main manuscript.
> >
> > We again thank the reviewer for the time and efforts for providing very helpful comments. If there are any more clarifications we could provide, we would be grateful if you could let us know.
> >
> > Best regards,
> >
> > Authors

---

> ### Comment · Reviewer_X8AG · 2021-08-26
> **Thanks for the clarification**
>
> Thanks for the clarification. I am convinced that this is a good paper, and hence my score of 7, accept!

---

### Decision · Program_Chairs · 2021-09-27

**Decision:**

Accept (Poster)

**Comment:**

This paper proposes a robust approach to dealing with stragglers and adversaries in federated learning by model grouping and staleness-dependent weighting of models at the server, combined with entropy-based filtering. Stragglers and adversaries are two key concerns in federated learning, and this paper makes progress along both fronts. The reviewers generally received this work well, and the authors’ responses helped address many of their concerns, leading reviewers to increase their scores. Overall, the consensus was that this paper is worthy of acceptance and makes a solid contribution to the federated learning literature. When preparing the camera ready, please make sure to address the questions that came up during the review process, especially around clarifying the proposed approach.